# Applied Clinical Tandem Mass Spectrometry-Based Quantification Methods for Lipid-Derived Biomarkers, Steroids and Cannabinoids: Fit-for-Purpose Validation Methods

**DOI:** 10.3390/biom13020383

**Published:** 2023-02-17

**Authors:** Isabelle Matias, Ilaria Belluomo, Pierre-Louis Raux, Monique Vallée

**Affiliations:** INSERM, Neurocentre Magendie, University of Bordeaux, U1215, F-33000 Bordeaux, France

**Keywords:** quantification methods, tandem mass spectrometry, steroids, cannabinoids, human plasma, GC-MS/MS, LC-MS/MS, validation

## Abstract

The emergence of metabolomics and quantification approaches is revealing new biomarkers applied to drug discovery. In this context, tandem mass spectrometry is the method of choice, requiring a specific validation process for preclinical and clinical applications. Research on the two classes of lipid mediators, steroids and cannabinoids, has revealed a potential interaction in cannabis addiction and metabolism-related disorders. Here we present the development of GC-MS/MS and LC-MS/MS methods for routine quantification of targeted steroids and cannabinoids, respectively. The methods were developed using an isotopic approach, including validation for linearity, selectivity, LLOQ determination, matrix effect, carryover, between- and within-run accuracy and precision, and stability tests to measure 11 steroids and seven cannabinoids in human plasma. These methods were satisfactory for most validity conditions, although not all met the acceptance criteria for all analytes. A comparison of calibration curves in biological and surrogate matrices and in methanol showed that the latter condition was more applicable for our quantification of endogenous compounds. In conclusion, the validation of our methods met the criteria for GLP-qualified rather than GLP-validated methods, which can be used for routine analytical studies for dedicated preclinical and clinical purposes, by combining appropriate system suitability testing, including quality controls in the biological matrix.

## 1. Introduction 

In preclinical and clinical research, the emergence of the term “omics” has spread in recent years to many multidisciplinary fields ranging from genomics, epigenomics, transcriptomics, and proteomics, as well as their downstream pathways that are embedded in the metabolomics field, one of the newest “omics” fields [1]. Therefore, characterizing metabolomic profiles can be a powerful tool for understanding the mechanisms that contribute to physiological and pathological phenotypes in a single individual. Namely, metabolomics provides the qualitative and quantitative analysis of endogenous small molecules present in different biological compartments, such as cells, tissues, organs, or biofluids, in response to environmental fluctuations or perturbations [2,3,4]. Hence, the content of metabolites fulfils the role of a central control tower of the organism and can account for the health status of individuals, whether in human or animal models. The resulting broader potential knowledge of biological mechanisms can accelerate preclinical and clinical research discoveries. Still, progress in this field is strictly linked to the evolution of analytical technologies used to identify and measure the levels of small molecules. 

Among many approaches available to measure a wide range of small molecules, immunoassays (IAs) represent the classic analytical techniques commonly used in clinical practice. This type of screening requires a simple process with minimal cost, and robotic techniques give the possibility of full automation. Despite this technique being easy to perform and effective in evaluating biomolecule concentrations, it still presents many limitations. First, ligand-binding assays are not available for all biomolecules. Second, when available, the antigen–antibody reaction presents possible crossover between similar molecules, therefore reducing specificity in many cases and leading to potential false-positive results [5,6,7,8]. In addition, since IAs target mainly one biomolecule, there is a lack of information about related metabolites which may have different molecular and/or cellular targets than the molecule of interest. Consequently, this methodology does not support the evaluation of pathways and networks in the same sample. 

To overcome these challenges, mass spectrometry (MS)-based technologies have been developed and have challenged the analytical chemistry field, becoming an essential tool in metabolomics [9,10]. MS techniques can ensure the assessment of several molecules in a single biological sample during the same analytical run, giving the possibility to evaluate a whole panel of compounds and pathway mapping. Additionally, chromatographic separation, often coupled to MS, assures the separation for the detection and measurement of similar molecules, such as isomers [11]. Another advantage of this analytical technique is the use of internal standards (ISs), providing high validity in terms of accuracy, precision, and reproducibility [12]. Although the non-targeted metabolomics hype aims to discover a large number of new molecules, metabolomics applied to targeted metabolites can provide a more appropriate tool for routine individual clinical diagnosis [13].

Lately, the development of tandem MS has offered a higher specificity together with better sensitivity and specificity [13]. This type of MS is considered the gold standard for targeted measurement of biomolecules when a data-driven panel of compounds has been established and quantitative methods can be designed ad hoc. Validation of the analytical methods is the key feature to ensure the robustness of every bioanalytical approach. Accordingly, tandem MS targeted method validation is a meticulous process aimed to achieve the best qualitative and quantitative performance for the measurement of an established group of compounds through the optimization and validation of different variables. This process is detailed in specific guidelines released and continuously revised by the European Medical Agency (EMA) in Europe and the Food and Drug Administration (FDA) in the United States [14,15]. These guidelines can be applied and adapted to different groups of metabolites.

Among the different biological classes of metabolites, lipids and lipid-derived compounds have attracted much interest in recent years because of their relevance in translational research and clinical applications [16,17,18]. Specifically, endogenous steroids (STs) and cannabinoids (CBs) can be identified as major targets and biomarkers for various neuroendocrine and behavioral-based dysfunctions [19,20,21,22,23,24]. For instance, our group and collaborators previously showed a link between these two classes of compounds in cannabis-related addiction [25] and energy balance-related disorders [26,27,28]. Our recent findings have highlighted the role of endogenous steroids, particularly pregnenolone (PREG), as an endogenous modulator of the type-1 cannabinoid (CB1) receptor [25,29], which is one of the main central targets that drive the bioactivity of cannabinoids, including the endocannabinoids anandamide (AEA) and 2-arachidonoylglycerol (2-AG), as well as the exogenous cannabinoid Δ^9^-tetrahydrocannabinol (THC), the main psychoactive component of *Cannabis sativa,* and its psychoactive metabolite 11OH-THC [30,31]. In addition, crosstalk between endogenous steroids (or neurosteroids when synthesized de novo within the brain) and endocannabinoids is an underlying pattern in cannabis addiction [32], as well as in metabolic-related disease [26,27,28] and stress-related disorders [22,33,34]. The growing interest in these compounds in preclinical and clinical research has led to the need to devise methods for their accurate, sensitive, and specific quantification. This is notably what has been addressed with tandem MS coupled with chromatographic techniques [6,13,35,36,37], although a few simultaneously analyzed endogenous and exogenous cannabinoids [38,39], as well as steroids and (endo)cannabinoids [40,41,42] in the same samples. 

For all these reasons, our research strategy was focused on targeted metabolomics involving the study of the two lipid subtypes, STs and CBs, for application-based research on potential biomarkers and drug discovery (Figure 1). 

Therefore, we have locally implemented two parallel quantification methods in our analytical platform for the quantification of STs and CBs involving highly specialized staff. In this paper, we present the development of two tandem MS methods, a gas chromatography–tandem mass spectrometry (GC-MS/MS) method for the measurement of eleven STs, including pregnenolone (PREG) and downstream steroids (Figure 2A), and liquid chromatography–tandem mass spectrometry (LC-MS/MS) method for the measurement of seven CBs, of which the two endocannabinoids AEA and 2-AG, and the related fatty acids, OEA (oleoylethanolamine) and PEA (palmitoylethanolamide) (Figure 2B), as well as THC together with its hydroxyl and carbonyl metabolites (11OH-THC and 11COOH-THC, respectively) (Figure 2C). 

Method development is an overall procedure that also includes pre-analytical and analytical phases that require a validation process before being applied to real samples as described in Figure 3. Once all the pre-analytical and analytical settings have been optimized, validation is a necessary step to define the limit of acceptability of the methods. In this paper, we focused on the development and validation of mass spectrometry methods that can be applied for analytical studies of molecules of research interest involving interconnected steroid and cannabinoid systems. Since sample extraction and chromatographic conditions have been previously established with good recovery and good separation of the analytes of interest [20,25,43,44,45], our work addressed the validation of mass spectrometric conditions, including ionization and fragmentation, which are important conditions governing a quantification method.

The choice of these methods meets local routine research needs for preclinical and clinical trials and the compounds included in the methods were selected based on their clinical and biological relevance. Our methods allowed the separation of similar compounds with specific biological functions and achieved relatively good results in terms of sensitivity and specificity for some of the targeted compounds. 

## 2. Methods 

Isotope dilution GC-MS/MS and LC-MS/MS methods were developed in parallel for the bioassay of clinically relevant STs and CBs in human plasma. All the experimental procedures were conducted on the basis of good laboratory practice-like (GLP-like) procedure. Indeed, the tests were conducted with reference to and adapting the guidelines for bioanalytical method development [14,15]. Precision and accuracy acceptability were established at CV < 20% and ±20%, respectively, which can be accepted criteria for mass spectrometry-based quantification of endogenous small molecules [46]. For calibration curves (except for LLOQ), the criteria were set at CV < 15% and ± 15% [46]. 

The following variables were evaluated for the validation of the two analytical methods: linearity, selectivity, determination of lower limit of quantification (LLOQ), matrix effect, carry-over, and within- and between-run determination of accuracy and precision. In addition, stability tests for MS/MS methods were performed with the stability during the preparation of samples in the biological matrix and post-preparative stability during runs on auto-sampler, as well as the stability in the solvent at room temperature (RT) and during storage conditions (−20 °C) of reference standards and deuterated analogues solutions, and finally, freeze and thaw stability tests were performed in biological matrix. 

For the method development, validation (Val), quality control (QC), calibration, and stability (Stab), samples were analyzed in the same analytical sequence. The Val samples, corresponding to samples of the neat matrix (methanol), were used to validate the accuracy and precision of the bioanalytical methods. These variables were evaluated by calculating the accuracy and precision of QC repeated measures. Calibration samples were used to construct calibration curves (CCs) from which the concentration of analyte in samples was determined. Finally, Stab samples were used to evaluate stability data in the matrix.

Recovery tests were not assessed for every sample preparation step since the quantification was performed using the isotope dilution method with the addition of stable deuterated isotopes as internal standards (ISs) at the beginning of the preparation process, which allows adjustment for any losses during sample preparation and to achieve accurate quantification. This method was performed using deuterated analogues of each analyte, except for the steroid EPIPRAG, which was quantified using the deuterated analogue of its isomer, PRAG (PRAG-d4). Each deuterated IS was differing from its analogues by a minimum of 3 mass units to avoid isotopic overlap during MS analysis [47]. The number and position of the deuterium atoms of the ISs are listed in Table 1. The source (supplier) and purity of each reference standard are indicated in Appendix A. 

### 2.1. Chromatographic and Mass Spectrometer Conditions

The chromatographic and ionization conditions, which are important clues of the method development, have been previously described [25,26,43,44,45,48].

For steroids, GC was operated using a 15 m Rtx-5Sil MS W/Integra Guard capillary column (Restek, Lisses, France) with a 0.25 mm inside diameter and 0.1 μm film thickness was employed for STs resolution. Injections were in splitless mode using helium as the carrier gas at constant flow at 1.2 mL/min, with the interface temperature at 290 °C. Optimization of the ionization conditions led to the use of the emission current at 30 µA, the source temperature at 210 °C, the methane at 2 mL/min as the reacting gas, and the argon as the collision gas. The ramp temperature has been optimized as followed: the initial GC temperature was 160 °C (1.25-min hold), followed by a temperature program to 230 °C at 50 °C/min, then to 260 °C at 4 °C/min, then to 290 °C at 50 °C/min, and finally to 320 °C at 5 °C/min, where it was held for 1 min. Mass spectra analysis of STs was acquired with a GC-MS/MS-XLS Ultra (ThermoElectron SAS, Villebon-sur-Yvette, France) operated in negative chemical ionization (NCI). NCI is a widely used ionization technique for GC-MS typically for analyzing small electrophilic molecules, as steroids, after derivatization of the analyte with chemicals that add electronegative atoms to the molecules. NCI is considered a soft ionization technique yielding a mass spectral pattern with less fragmentation in which the molecular or pseudo-molecular ions are easily identified. NCI shows a dominant molecular ion peak (Mˉ˙) as we determined for all STs compounds in full-scan MS experiments over a range of 100 < *m*/*z* < 800 (Table 2).

For cannabinoids, the chromatographic separation of CBs was achieved using a C18 Discovery column (5 µm, 15 cm × 4.6 mm; Supelco, Saint-Quentin-Fallavier, France) with a guard column using an isocratic gradient (85% methanol, 15% water, 0.1% formic acid). For the mobile phase, methanol and acetonitrile combined with water are commonly used. These solvents were tested, and we found that methanol provided better ionization and less ion suppression than acetonitrile for the majority of the target analytes. To increase ionization, the addition of organic modifiers has been reported. Formic acid, acetic acid, and ammonium acetate were tested in the mobile phase, resulting in higher ion intensity with formic acid under our conditions. Different methanol gradients were then tested, and the isocratic mode allowed a better separation for our analytes, which have a very close polarity. The optimal HPLC conditions, column temperature, gradient composition, and pH were selected based on the resolution and intensity of each peak. The resulting retention times are shown in Table 2 for each compound. Mass spectra analysis of CBs was acquired with an LC-MS/MS TSQ Quantum Access triple quadrupole instrument (ThermoElectron SAS, Villebon-sur-Yvette, France) equipped with an APCI (atmospheric pressure chemical ionization) source and operating in positive ion mode. The source conditions were 350 °C for the capillary and the vaporizer temperatures, 10 mV for the discharge current, 35 mTor for the sheath gas pressure, and 10 mTor for auxiliary gas pressure, with an argon collision gas pressure set up at 1.5mTor. The protonated molecular ions for all CBs were determined in full-scan MS experiments over a range of 50 < *m*/*z* < 500 by triple-quadrupole APCI+. The protonated molecular ion [M + H]^+^ and the most intense adduct ion, for each standard, are showed in Table 2. For 2-AG, we obtained two peaks (as shown in Appendix A) corresponding to 2-AG and its inactive isomeric form 1-AG that appears upon extraction. For quantification, we summed the concentrations of both isomers providing meaningful data for biological interpretation, as is commonly performed [49,50].

The tandem mass spectrometer settings of our method development allowed the selected reaction monitoring (SRM) mode process that was operating to enhance sensitivity. For both STs and CBs, the ions (precursor and product) corresponding to each analyte and IS were then identified through analysis of authentic pure reference standards (>95% purity, as referred in Appendix A) dissolved in methanol, to check the specificity (i.e., non-contamination) of the solution, and determine retention time and MS conditions. Precursor and product ions of the highest intensity for both STs and CBs, corresponding to the expected chemical structure following the positive or negative ionization, were chosen as representative of each analyte and IS included in the MS method. To assure the maximum production yield of the product ions, the collision energy (CE) was determined for each compound using a pre-set ramp of values, ranging from 10 to 30 eV. The precursor ion, product ion, collision energy (eV), and retention time of each analyte and IS included in the methods are listed in Table 2. Representative examples of the SRM chromatograms for both STs and CBs obtained from a standard mixture are shown in Appendix A, respectively.

### 2.2. Sample Preparation

Val, QC, CC, and Stab samples were prepared with working solutions of standard analytes spiked with ISs, which were obtained by dilution from separate stock solutions. The concentration of the stock and working solutions are listed in Appendix A. Each solution was prepared on the day of the analysis. Val, CC, and QC samples were prepared in methanol, and Stab samples were prepared in commercial plasma (human EDTA-3K plasma pool; PLA022; Dutscher SAS, Bernolsheim, France) and stored at −20 °C, before the evaluation of stability in the biological matrix. 

Three concentration levels were used for Val and QC samples: at low, mid, and high concentrations corresponding to the calibration curve level CC2 (approximately 2 times LLOQ), CC4 (8 times LLOQ), and CC8 (0.5 times ULOQ), respectively. LLOQ (low limit of quantification) and ULOQ (upper limit of quantification) samples were the calibration samples of the lowest and highest concentration in the CC, respectively, that can be quantified with acceptable (in our case within a 20% CV) accuracy, precision, and linearity within the curve [15]. In practice, the analysis of LLOQ results in a signal at least 10× the standard deviation of the blank sample [44], and can be also estimated based on the signal-to-ratio (S/N), with S/N ≥10 LLOQ [14]. 

For samples in matrices, STs and CBs extraction protocols were previously optimised [20,25,26,43,44,45,48]. Briefly, STs were extracted after homogenization of plasma with methanol/H_2_O (75/25, *v*/*v*) containing their respective deuterated ISs, and purified by a simple solid-phase (SPE) extraction method using reverse-phase C18 columns. Free steroid fraction was obtained by eluting the column with methanol. CBs were extracted by liquid–liquid extraction (LLE) with chloroform after homogenization of plasma with chloroform/methanol/Tris-HCl 50mM pH 7.5 (2:1:1, *v*/*v*) containing their respective ISs, and then purified by SPE-C18. CBs fractions were obtained by eluting the column with 1:1 (*v*/*v*) cyclohexane/ethyl acetate. 

Then, for both analyses, lipid extracts were concentrated on a nitrogen stream evaporator. The subsequent deconjugation and derivatization steps were performed to, respectively, release the free steroids and to increase volatility, heat resistance, and ionizability. The formation of pentafluorobenzyl oximes for NCI detection was followed by trimethylsilyl ether formation for adequate sensitivity and selectivity. Then, the derivatized ST samples and the dried CB samples reconstituted with methanol were transferred into autosampler vials for mass quantification.

## 3. Results

### 3.1. Calibration Curves and Linearity 

The calibration curves, consisting of ten points of calibration including the zero samples, LLOQ and ULOQ (Table 3), were obtained with increasing amounts of reference standards supplementing with the same amount of ISs according to the isotopic dilution method [51,52]. The ranges of the calibration curve were chosen to span across anticipated concentrations of the target endogenous analytes in real human samples, with an approximate average at the mid-level of the curve. This method used deuterated analogues of analytes as ISs (Table 1). The ratio of the peak areas of each analyte to its deuterated analogue was used for quantification, except for EPIPRAG, for which the ratio EPIPRAG/PRAG-d4 was used to calculate its concentration since any deuterated analogue of EPIPRAG was commercially available and both the chromatographic elution time and physicochemical properties were similar between EPIPRAG and PRAG. In the zero samples (CC0), only the known amount of ISs was added with no analyte of interest. Levels in the calibration curves were expressed as absolute amounts of reference standards. As such, the volume of the sample during extraction and/or injection can be adjusted for better sensitivity. For quantification, the ratio of peaks was plotted against a corresponding amount of the calibration curve, which was then normalized to the volume of each analyzed sample.

A first preliminary test method was performed to choose the most appropriate matrix by comparison of three calibration curves (in terms of slope, response, and recovery) in different matrices: methanol, surrogate matrix (PBS containing 6% of BSA), and biological matrix (human plasma). The surrogate matrix was used as an analyte-free matrix to mimic the targeted biological matrix. Slope, intercept, r2 difference, and recovery were compared to the methanol CC, used as a reference. CC points were accepted when the residual accuracy from the theoretical concentrations was between 75% and 115%, and a CC was accepted when at least 70% of the points had an accepted accuracy of the residuals. Methanol CCs, prepared in triplicate, showed a good performance in terms of accuracy and repeatability (within-day or intra-run precision). In surrogate and biological matrices, recovery assessment was not always optimal and resulted in some unacceptable accuracy values, particularly for compounds with high concentrations and/or low sensitivity. Thus, only 5 out of 11 STs and four out of seven CBs were validated in PBS/BSA, and 7 out of 11 STs and two out of seven CBs were validated in human plasma. In addition, given that the comparison test between CCs in each matrix and CCs in methanol was found to be suitable, and that CCs in methanol provided better accuracy than CCs in the matrices, we next tested the repeatability for linearity by comparing three additional CCs in methanol. The CC of the 11 STs and seven CBs met the following acceptance criteria: accuracy within ±15% of the theoretical concentration for each point, 70% of the calibration samples above acceptance criterion, the coefficient of correlation (R2) greater than 0.99, and the slope significantly different from zero (*p* > 0.05). Hence, methanol CCs were used in the following steps of the development of both methods.

### 3.2. Preparative Stability and Post-Preparative Stability at the Auto-Sampler Temperature 

The preparative stability was tested for the preparation steps for the GC-MS/MS and LC-MS/MS methods using three Stab samples at low and high concentrations (Stab low and Stab high, respectively). The aim was to assess whether the several steps of the sample preparation could be performed on successive days or on the same day (control condition). The mean of the endogenous concentration was calculated and subtracted from the concentration measured in each Stab sample. The mean of the concentrations observed at each experimental condition was compared to the control condition by calculating the% of the difference (i.e., coefficient of variation, CV) (results are shown in Table 4). Acceptability was set at ±20%. The preparative stability met acceptance criteria at high concentrations for the experimental conditions B (24 h at −20 °C following extraction) for all STs, but only for 4 out of 11 STs at low concentrations. At both concentrations, CVs < 20% were obtained for condition C (24 h at −20 °C following step 1 of derivatization), except for 17OH-PROG. For the CBs assay, the preparative stability met acceptance criteria for almost all the CBs at the three conditions: delay at −20 °C after purification (condition B), after extraction (condition C), or after extraction and purification (condition D). CVs < 30% were observed for OEA at low and high concentrations for conditions B and C, respectively, and CVs > 30% were found at low concentrations for OEA (condition D) and 11COOH-THC at the three conditions. According to these data, all steps in the preparation of experimental biological samples in routine ST and CB measurements should be conducted on the same day.

The post-preparative stability test consisted of evaluating the processed sample stability at the auto-sampler temperature at RT or 9 °C according to the GC-MS/MS or LC-MS/MS methods, respectively. Three Stab samples at two concentration levels (low and high) were assayed, and the same sample was assayed at T0 and at three periods of time: T1 (12 h), T2 (24 h), and T3 (36 h). The mean of the endogenous concentration was calculated and subtracted from the concentration measured in each Stab sample. For each period, the mean concentrations observed (n = 3) in Stab samples at each concentration level (low or high) was compared to the mean concentration observed at T0 (results are shown in Table 5). Acceptability was set at ±20%. The corresponding CVs of post-preparative stability were acceptable or close to acceptance for all CBs and STs at T1, except for 11OH-THC at low concentration. CVs < 20% were obtained for 10 out of 11 STs (CV > 30% for PROG) at low concentrations and for all STs at high concentrations at T2, and 9 out of 10 STs (CV > 30% for PREG and EPIALLO) at high concentrations and for all STs at low concentrations at T3. In addition, post-preparative stability met acceptance criteria for five out of seven CBs (20% < CV < 30% for OEA; CV > 30% for 11OH-THC) at low concentrations and for all CBs at high concentration at T2, and for five out of seven CBs (CV > 50% for OEA and 11OH-THC) at low concentrations and for six out of seven CBs (CV > 50% for OEA) at high concentration at T3. Overall, these data demonstrate that injection on the auto-sampler should be carried out within 12 h after extraction for both STs and CBs. 

### 3.3. Selectivity 

Although all deuterated ISs were commercially certified (at least 95% of purity, Appendix A), the deuterium stability was tested and the cross-contamination between compounds and their respective ISs was evaluated, calculating the percentage of the area of each compound in the ISs analyzed at working concentration. The concentration of each IS was chosen to obtain peak areas that could be measured with good accuracy and precision, corresponding to ~10-fold mid-concentration of the calibration curve. No H/D exchange was found for the ISs, except for 3α,5α-THDOC-d4 which lost one deuterium, resulting in the analysis of 3α,5α-THDOC-d3. The calculated% of analyte in IS solution was lower than 0.03% for six out of seven CBs and 2 STs (ALLO-d4 and DHT-d3) out of 11. It was lower than 0.9% for THC-d3, and the other STs, except for 3α,5α-THDOC-d3, for which new stock and working solutions were made to meet acceptance criteria. In addition, the percentage of ISs in the compound was analyzed at ULOQ. The calculated% ISs in the analyte solutions was between 0.02% and 0.8%, except for the steroid TESTO, for which new stock and working solutions, then reaching the acceptance criteria, were made. Thus, in our routine analysis, special attention was paid to selectivity, and new stock and working solutions were prepared as necessary. Therefore, in each run of human samples, ISs at the working concentration were analyzed in zero samples (blank samples spiked with ISs), to check the percentage of the compound (target analyte) in the IS solution.

### 3.4. Determination of the LLOQ 

LLOQ was determined as the smallest amount that could be quantified for each analyte with acceptable accuracy from the theoretical value (±20%) and precision (CV < 20%). LLOQ was corresponding to the lowest quantifiable point of the CC (CC1) (Table 3). 

### 3.5. Matrix Effect

The matrix effect was tested by comparing Val samples in the biological matrix and methanol at three concentration levels (low, mid, and high) with the acceptance criterion set within ±20%. The results are shown in Table 6. A good percentage of difference was calculated for almost all the STs and CBs at mid and high concentrations, although the % difference was above but very close to ±20% for PROG, EPIPRAG, and PRAG at the high level and for PEA at the mid-level. At low concentrations, only 2 STs (ALLO and TESTO) out of 11, and four (2-AG, PEA, THC, 11OH-THC) out of seven CBs reach good acceptability, mainly due to the presence of endogenous STs and CBs in the biological matrix. This may explain the better validation criteria obtained for CBs in methanol. 

### 3.6. Carry-Over 

Six blank samples in methanol were analyzed after ULOQ to evaluate carry-over. This experiment was repeated six times. The percentage of the area contained in the blank was calculated compared to the LLOQ area, using an acceptance criterion of ±20%. No carry-over was observed for 8 STs out of the 11 included in the method. For TESTO, ALLO, and EPIALLO, carry-over in the first blank was observed only in one ULOQ analysis out of six. Carry-over was found for five out of seven CBs (it was absent for 11COOH-THC and 11OH-THC) in the first blank sample analysis, while it was absent in the following five blanks.

As a result, we decided in our method to add three methanol blank injections in the analytical run following the ULOQ sample, corresponding to the highest point of the CC (CC10), and one methanol blank sample following CC6, CC7, and CC8. These blank injections were used in all the analytical runs for human sample quantification. In the biological samples, no carry-over was observed in the washing vial (with pure methanol) used to clean the syringe before the injection of each sample. 

### 3.7. Within and between Run Accuracy and Precision 

Accuracy and precision were determined within and between runs, to assess the repeatability and degree of closeness to the theoretical values of the measurements. Accuracy represents the evaluation of the measurement compared to the true value [12,14,15]. It was assessed by calculating the deviation of a predicted concentration from its nominal value. The precision of the analytical system is a measure of the repeatability of instrument performance determined by repetitive injection of the same sample. Three concentrations were evaluated for each compound: low, mid, and high concentrations. Three different analytical sequences were analyzed on three different days (n = 5 per run and per concentration for each sequence). The within-run accuracy was calculated with the% error taking into account the five replicates of each run, and the between-run accuracy was calculated with the between-run% error taking into account the 15 replicates of the three runs. The within-run precision was evaluated with the% within-run coefficients of variation (CV) using the within-group mean square (WMS) value obtained from the ANOVA. The between-run precision was evaluated with the% between-run coefficients of variation using the between-group mean square (BMS) and WMS values obtained from the ANOVA. Accuracy and precision acceptability were established at CV < 20% and ±20%, respectively. Accuracy and precision results for both STs and CBs are shown in Table 7 and Table 8, respectively. Overall, STs showed good accuracy and precision, except at low levels for 17OH-PROG, EPIPRAG, DHEA, TESTO, and DHT that display some CV in the range of 30–50%, and for EPIALLO, PROG, and 3α,5α-THDOC, which had a CV > 50%. In addition, PROG accuracy was not acceptable at the mid-concentration three times out of four. Values of both accuracy and precision were acceptable or close to the acceptance threshold for all the CBs, except 11COOH-THC accuracy at low concentration and both accuracy and precision for OEA at all levels. 

### 3.8. Solution Stability in the Solvent at RT and during Storage at −20 °C

The stability of compounds was checked for both stock solutions (StSs) and working solutions (WSs) of analytes and ISs. StSs corresponded to the most concentrated standard or IS solutions, purchased from a specialized commercial supplier, from which stock solutions and working solutions are made (Appendix A). StSs were purchased as powders, with a known referenced weight, and then dissolved in a solvent or purchased as a liquid solution with a known referenced concentration. The stability was evaluated for freshly made WSs at room temperature for 4 h to mimic the time they are on the bench during the preparation of a CC (usually not more than two hours, but it was considered in excess). For both WSs and StSs, the stability was calculated after 3 months at −20 °C. This would be the maximum storage time between each preparation of fresh solutions. Results, calculated as a mean% difference between a fresh solution and an evaluated solution in terms of mean area ratio (analyte area/IS area) are shown in Table 9. Acceptability was set at ±20%. Stability was good after four hours at room temperature for all the compounds, except for EPIPRAG and TESTO, which had 20% < CV < 30%. Following the 3 months at −20 °C, the mean% difference values for StSs and WSs met acceptability for almost all the compounds. A 20% < CV < 30% was observed for the StSs of PROG, ALLO, TESTO, AEA and the WSs of 17OH-PROG, EPIPRAG, 2-AG, AEA, and OEA. In addition, the StS of OEA and WS of PROG had a CV > 30%. Hence, in our methods for human samples, StSs and WSs were prepared within a shorter time frame to meet the acceptance criteria. 

### 3.9. Freeze and Thaw Stability

The freeze and thaw stability tests were assessed with three freeze/thaw cycles (T0, T1, T2, T3) with a 24 h-delay between each of them for four Stab samples at middle and high concentrations. This test aimed at assessing how many times a biological matrix sample (plasma) could be successively thawed without interfering with analyses. The mean percentage of the difference between the analyzed time point and time zero (T0) was calculated. The results are listed in Table 10. Our results clearly showed that thawing a sample for the third time was altering the measurement of almost all the compounds, except for TESTO, DHT, and 11COOH-THC. Results for high concentrations were good at the second cycle for all the compounds, while at medium concentration some of the STs presented levels lower than T0. A single freeze/thaw cycle did not compromise the accuracy of the measurement for any compounds at all concentrations, except for 3α,5α-THDOC. Therefore, we strongly advise and ensure that the cold chain of samples should not be interrupted during sample preparation and storage for ST and CB assays.

## 4. Discussion

The present work evaluated six validation variables and five stability tests for measuring targeted steroids (STs) and cannabinoids (CBs) using isotopic dilution-based tandem mass spectrometry (GC-MS/MS and LC-MS/MS, respectively) in individual human plasma samples. Both methods described here meet most, yet not all, performance capabilities for several specifications, including linearity, selectivity, specificity/carryover, accuracy, and precision. 

The development of our methods showed that calibration curves (CCs) in methanol were more compliant with the quantification of STs and CBs analyzed in human plasma samples. While it is preferably recommended to perform CCs in the biological matrix for GLP method validation for exogenous compounds according to guidelines [14,15], our methods with CCs in methanol prevent the presence of endogenous STs and CBs in biological matrices. An alternative could be proposed with the use of charcoal-stripped plasma as a matrix presumably devoid of endogenous compounds, but in our experience, a large variability in the results occurs with the use of this type of matrix, due to the presence of residual endogenous compounds and/or when spiked analytes bind to remaining traces of charcoal [53].

Although our methods are not strictly considered as GLP-validated methods according to guidelines [14,15], they meet the requirements for GLP-qualified methods [54,55,56,57]. This implies in particular that the validation process can be defined in terms of “fitness for purpose”, depending on the scope of the method, and can be performed at different steps of an application [58,59]. In large part, few validation methods strictly follow specifications and quality assurance, and can achieve all acceptance criteria for all analyzed compounds. A compromise is, therefore, appropriate depending on the type of investigation performed. In addition, for qualified methods, operational elements can be implemented to counterbalance some of the method’s weaknesses. In this regard, it is worth considering the high variability of the results we obtained, especially at low concentration levels for the quantification of specific endogenous compounds in real samples. An additional validation process of our qualified methods would be required for the quantification of these compounds, especially if low concentrations are expected. Moreover, the LLOQ of some compounds do not meet acceptance criteria in terms of accuracy and precision, so the values could be higher and determined more thoroughly. This might be due to endogenous levels that are higher than the spiked amount used for the LLOQ. 

In addition, in other to monitor the validity of our qualified methods over time and across projects, we included in the daily run of the methods, appropriate mass spectrometer maintenance, calibrated instrumentation testing, and system suitability criteria testing, using sufficient blanks to avoid contamination and carry-over, as well as quality controls (QCs). QC samples were prepared in matrices matching the biological samples to overcome the matrix interference effect. QC samples were also used to assess method accuracy and precision to ensure the integrity of the assay for each analytical run of our qualified methods. In addition, we developed quality assurance procedures, including standard operating procedures (SOPs) for reliable quantification of STs and CBs. This allows the evaluation of acceptable assay performance for each specific analytical application, taking into account method deviations that may occur and adjusting the method accordingly for approval. All these features contribute to the validity of our qualified methods that can then be used for daily analysis by highly specialized staff. Consequently, our methods follow the operational controls required for the qualification of analytical methods involving a context of use, i.e., the suitability of methods for research applications [60]. The routine workflow of our methods for biological samples is described in Figure 4.

## 5. Conclusions

Our methods meet the stated criteria for recurrent measurement of STs and CBs in human plasma and saliva, as well as in plasma and tissue samples (small brain areas, tissues, and organs) from animal models (Figure 3). The application of both methods has, therefore, been performed in several preclinical and clinical projects. As such, spectrometry-based studies of target steroid metabolites in animal models of cannabis addiction have highlighted the crucial regulatory function of the (neuro)steroid pregnenolone (PREG) in the toxic effects of cannabis, mimicked by the action of THC on the CB1 receptor [25]. Thus, the analytical data supported the discovery of the PREG-CB1 regulatory loop [32], and are the basis for the development of a translation project in humans. Consequently, both analytical methods are currently used to assess secondary outcome measures in clinical trials related to cannabis use disorders (www.clinicaltrial.gov (accessed on 8 February 2023)*,* NCT03325595, NCT03443895, NCT03717272, NCT05451017). In addition, these methods have been applied to research projects involving endocannabinoids and pregnenolone in the control of energy balance in post-obese patients (CannaPreg project, ERC grant 640923) [26], as well as in type 2 diabetes (PREVIEW cohort; www.clinicaltrial.gov, NCT01777893) [27,28].

Nevertheless, our methods have some limitations and future development would be beneficial. First, other STs and CBs of interest could be added to our analyses. For instance, we are in the process of analyzing 17OH-PREG with our GC-MS/MS method for rodent plasma samples, since the delta 5 (involving PREG metabolism to 17OH-PREG) and delta 4 (involving PREG metabolism to PROG and then to 17OH-PROG) pathways are predominant in rodents and humans, respectively [61]. We also plan to supplement our LC-MS/MS method with the assay of the phytocannabinoids of increasing clinical interest, cannabidiol (CBD) and cannabinol (CBN), which both increase in plasma after smoking cannabis cigarettes but are much less psychoactive than THC [30,62]. In addition, future development on sulphated steroids could be valuable in light of emerging research on dysfunctions in the equilibrium between unconjugated and conjugated (sulphated) steroids involved in the pathophysiology of several diseases [63]. Finally, methods could be developed for future research projects based on a mixed untargeted and targeted approach, as described for carbonyl steroid profiling analysis using an LC-MS/MS method [64].

## Figures and Tables

**Figure 1 biomolecules-13-00383-f001:**
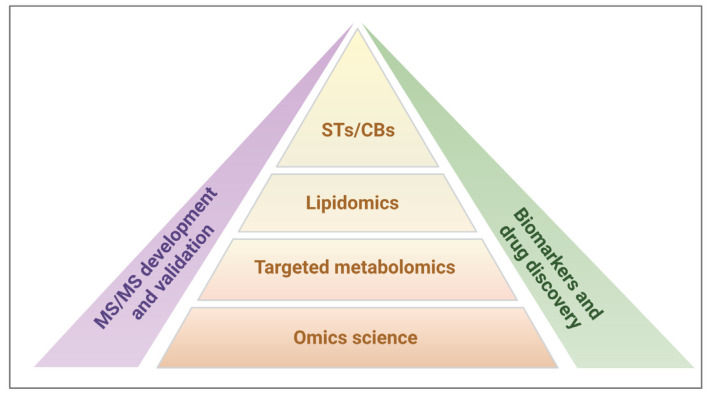
Schematic representation of our analytical research strategy to assess the cross-functional interplay between steroid and cannabinoid systems. Among the omics research topics, our targeted metabolomics approach focuses on lipidomics, specifically on steroids (STs) and cannabinoids (CBs). Our studies benefit from advances in the development and validation of tandem mass spectrometry (MS/MS) methods to provide new insights into biomarkers and drug discovery.

**Figure 2 biomolecules-13-00383-f002:**
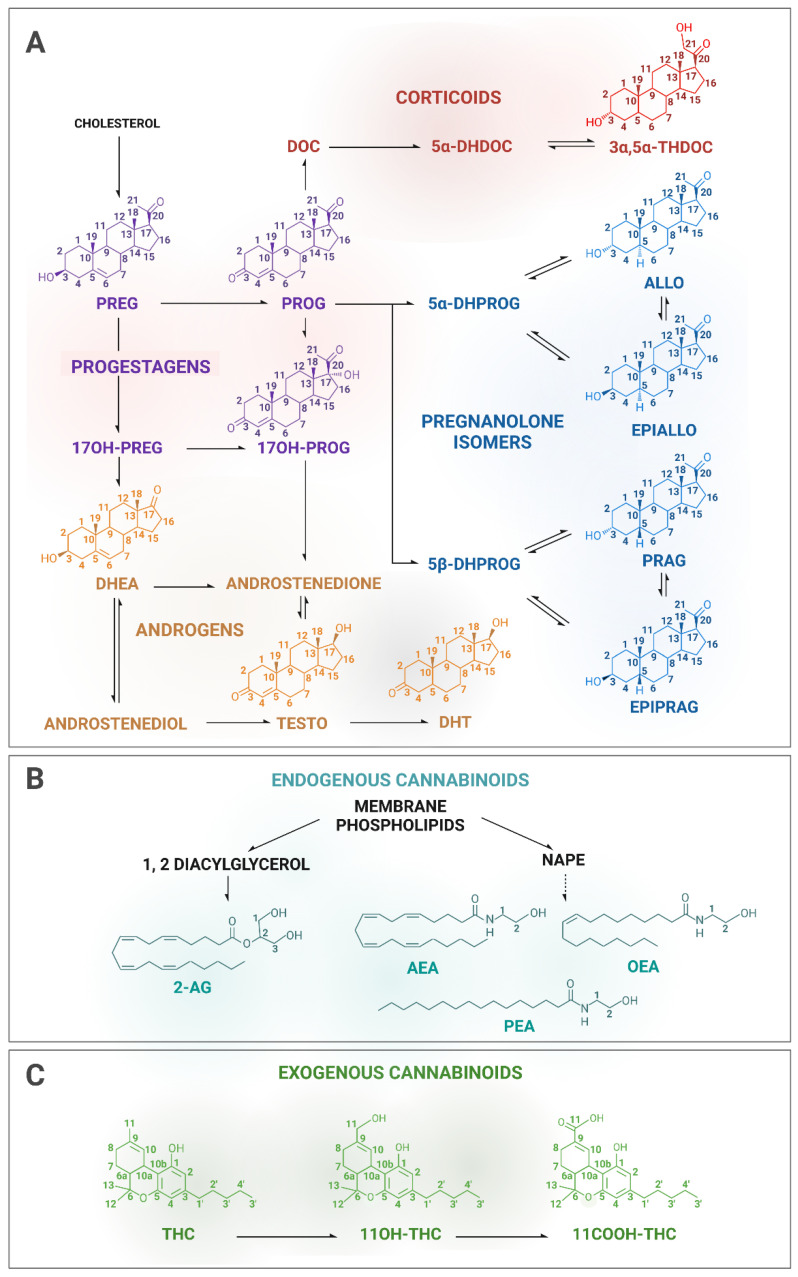
Structures and metabolic pathways of the targeted molecules. The steroids of interest include four progestagens (PREG, PROG and its 17hydroxy metabolite), one corticoid (3α,5α-THDOC), four pregnanolone isomers (ALLO, EPIALLO, PRAG and EPIPRAG), and two androgens (TESTO and DHT) (**A**). The cannabinoids of interest include two endocannabinoids (AEA and 2-AG), and two related fatty acids (OEA and PEA) (**B**) and the exogenous cannabinoid THC and its two metabolites (11OH-THC and 11COOH-THC) (**C**). Not all metabolic pathways are represented. See abbreviations in the abbreviation section.

**Figure 3 biomolecules-13-00383-f003:**
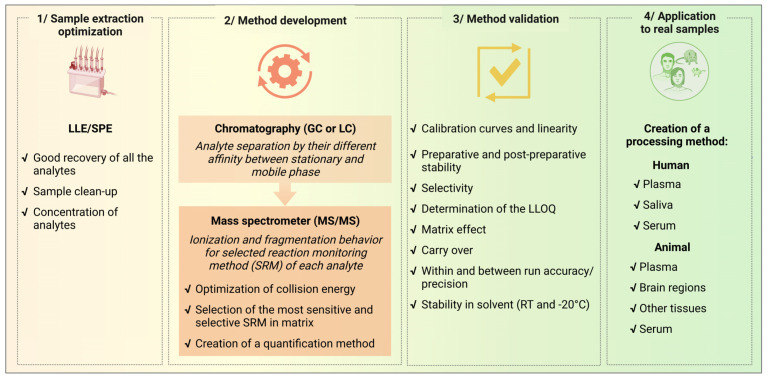
**Pre-analytical and analytical phases of the validation process.** The steps of the methods include 1/sample extraction optimization, 2/mass spectrometry method development, 3/the validation process of the method, and 4/the application to real biological samples in human and animal models.

**Figure 4 biomolecules-13-00383-f004:**
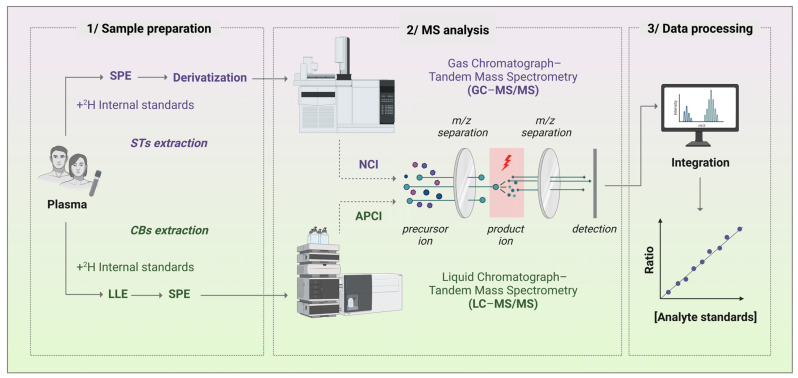
**Routine analytical workflow in biological samples.** 1/Sample preparation involved the addition of deuterated (^2^H)-labelled internal standards (ISs) to samples prior to solid phase extraction (SPE) and derivatization processes for steroids (STs) or prior to liquid–liquid and solid phase extraction (LLE and SPE) steps for cannabinoids (CBs). 2/NCI GC-MS/MS and ACPI LC-MS/MS analyses were performed for STs and CBs, respectively. Both tandem MS included *m*/*z* separation for selective precursor and product ions of each analyte for detection. 3/Data processing required peak integration of the targeted analytes and their respective ISs, and quantification based on an analyte calibration curve obtained from the analyte-to-IS ratio of peak intensities for increasing concentrations of analyte standards.

**Table 1 biomolecules-13-00383-t001:** Position of the deuterium atoms of the internal standards (ISs) as deuterated analogues of the STs and CBs of interest.

**IS—STs**	**C Positions**
PREG-d4	7,21,21,21
PROG-d9	2,2,4,6,6,17α,21,21,21
17OH-PROG-d8	2,2,4,6,6,21,21,21
ALLO-d4	C 17,21,21,21
EPIALLO-d4	17α,21,21,21
PRAG-d4	17α,21,21,21
3α,5α-THDOC-d4	17,21,21
!DHEA-d5	2,2,3,4,4
TESTO-d3	16,16,17
DHT-d3	16,16,17
**IS—CBs**	**Positions**
2-AG-d5	1, 1, 2, 3, and 3 positions of the glycerol moiety
AEA-d4	hydroxyethyl 1,1,2,2
OEA-d4	hydroxyethyl-1,1,2,2
PEA-d4	hydroxyethyl-1,1,2,2
THC-d3	3 deuteriums at the C5 position of the pentyl chain
11COOH-THC-d3	3 deuteriums at the C5 position of the pentyl chain
11OH-THC-d3	3 deuteriums at the C5 position of the pentyl chain

**Table 2 biomolecules-13-00383-t002:** Precursor ion, product ion, collision energy (eV), and retention time of each analyte and IS of STs (scan time 0.1s, dwell time 1 ms, scan width 0.8) and of CBs (scan time 0.1 s; dwell time 2 ms, scan width 0.01).

**STs**					
**Class**	**Analyte and IS**	**Precursor Ion (*m*/*z*)**	**Product Ion (*m*/*z*)**	**Collision Energy (eV)**	**Retention Time (min)**
**PROGESTAGENS**	PREG	405.3	272.4	15	9.8
	PREG-d4	409.4	273.5	15	9.8
	PROG	684.4	654.6	10	13.5
	PROG-d9	693.5	663.7	10	13.4
	17OH-PROG	461.2	411.4	20	8.6
	17OH-PROG-d8	466.5	415.4	20	8.5
**PREGNANOLONE ISOMERS**	ALLO	407.3	274.2	15	9.0
	ALLO-d4	411.4	275.5	15	9.0
	EPIALLO	407.3	274.5	15	9.9
	EPIALLO-d4	411.4	275.5	15	9.9
	EPIPRAG	407.4	274.5	15	8.8
	PRAG	407.3	274.2	15	9.1
	PRAG-d4	411.3	275.2	15	9.1
**CORTI COID**	3α,5α-THDOC	493.1	88.9	15	10.1
	3α,5α-THDOC-d3	496.2	88.9	15	10.1
**ANDROGENS**	DHEA	535.3	215.7	20	8.0
	DHEA-d5	540.5	216.0	25	8.0
	TESTO	535.3	413.7	20	8.7
	TESTO-d3	538.3	416.7	20	8.6
	DHT	537.3	190.5	25	8.2
	DHT-d3	540.5	190.9	25	8.2
**CBs**					
**Class**	**Analyte and IS**	**Precursor Ion (*m*/*z*)**	**Product Ion (*m*/*z*)**	**Collision Energy (eV)**	**Retention Time (min)**
**ENDOCANNABINOIDS**	2-AG /1-AG	379.2	287.0	14	5.8/6.2
	2-AG-d5 /1-AG-d5	384.2	287.0	16	5.8/6.2
	AEA	348.2	287.1	13	5.5
	AEA-d4	352.2	287.1	14	5.5
	OEA	326.2	62.2	18	7.6
	OEA-d4	330.3	66.6	16	7.6
	PEA	300.2	62.4	16	7.0
	PEA-d4	304.2	66.5	15	7.0
**THC and METABOLITES**	THC	315.2	193.0	22	5.3
	THC-d3	318.2	196.1	23	5.3
	11COOH-THC	345.2	327.1	14	3.4
	11COOH-THC-d3	348.2	330.2	14	3.4
	11OH-THC	331.2	313.1	12	3.1
	11OH-THC-d3	334.2	316.0	14	3.1

**Table 3 biomolecules-13-00383-t003:** Calibration curve levels (from CC0 to CC9) expressed as absolute amounts of reference standards for steroid (STs) (**A**) and cannabinoid (CBs) (**B**) assays.

(**A**) **STs**											
**Analyte**	**Unit**	**CC0**	**CC1**	**CC2**	**CC3**	**CC4**	**CC5**	**CC6**	**CC7**	**CC8**	**CC9**
PREG	pg	0	0.5	1	2	4	8	16	32	160	320
PROG	pg	0	25	50	100	200	400	800	1600	8000	16,000
17OH-PROG	pg	0	0.025	0.05	0.1	0.2	0.4	0.8	1.6	8	16
ALLO	pg	0	0.5	1	2	4	8	16	32	160	320
EPIALLO	pg	0	0.5	1	2	4	8	16	32	160	320
EPIPRAG	pg	0	0.025	0.05	0.1	0.2	0.4	0.8	1.6	8	16
PRAG	pg	0	0.025	0.05	0.1	0.2	0.4	0.8	1.6	8	16
3α,5α-THDOC	pg	0	0.5	1	2	4	8	16	32	160	320
TESTO	pg	0	0.5	1	2	4	8	16	32	160	320
DHEA	pg	0	0.5	1	2	4	8	16	32	160	320
DHT	pg	0	0.5	1	2	4	8	16	32	160	320
(**B**) **CBs**											
**Analyte**	**Unit**	**CC0**	**CC1**	**CC2**	**CC3**	**CC4**	**CC5**	**CC6**	**CC7**	**CC8**	**CC9**
2-AG	pmol	0	0.75	1.5	3	7.5	15	30	75	150	300
AEA	pmol	0	0.075	0.15	0.3	0.75	1.5	3	7.5	15	30
OEA	pmol	0	0.75	1.5	3	7.5	15	30	75	150	300
PEA	pmol	0	0.75	1.5	3	7.5	15	30	75	150	300
THC	ng	0	0.75	1.5	3	7.5	15	30	75	150	300
11COOH-THC	ng	0	0.75	1.5	3	7.5	15	30	75	150	300
11OH-THC	ng	0	0.75	1.5	3	7.5	15	30	75	150	300

pmol to pg conversion factor: pg = pmol × MW. MW (g/mol): 2-AG: 347.5; AEA: 378.5; OEA: 325.5; PEA: 299.5.

**Table 4 biomolecules-13-00383-t004:** Preparative stability for steroids (STs) (A) and cannabinoids (CBs) (B) expressed as mean% difference from the control condition for Stab samples (n = 3) at low and high concentration levels [C] (corresponding to CC2, CC8 concentration levels, respectively, of the calibration curve). The occurrence of a 24 h time interval at −20 °C between the successive extraction, purification, and injection steps was tested. In condition B, extraction and purification were performed on the same day, and injections were delayed by 24 h. In conditions C and D, extraction and purification were separated by 24 h, and then injection was performed consecutively to purification or postponed for 24 h in conditions C and D, respectively.

**A. STs**						
	**CONDITION B**		**CONDITION C**			
**Analyte**	**Low [C]**	**High [C]**	**Low [C]**	**High [C]**		
PREG	32.6%	−3.1%	5.8%	−3.7%		
PROG	75.4%	12.5%	6.7%	−3.2%		
17OH-PROG	50.4%	11.5%	−55.6%	−37.5%		
ALLO	73.2%	−1.2%	7.6%	−11.0%		
EPIALLO	44.1%	−0.4%	1.6%	0.2%		
EPIPRAG	36.7%	−1.1%	23.4%	−0.9%		
PRAG	82.5%	4.5%	22.8%	−2.7%		
3α,5α-THDOC	−3.7%	13.7%	−15.3%	15.6%		
DHEA	−6.2%	2.8%	−12.2%	2.8%		
TESTO	5.2%	−8.7%	5.5%	−8.5%		
DHT	4.3%	3.4%	2.4%	2.6%		
**B. CBs**						
	**CONDITION B**		**CONDITION C**		**CONDITION D**	
**Analyte**	**Low [C]**	**High [C]**	**Low [C]**	**High [C]**	**Low [C]**	**High [C]**
2-AG	9.2%	2.4%	15.4%	3.8%	7.6%	3.9%
AEA	12.2%	0.4%	8.9%	1.3%	6.9%	5.0%
OEA	−30.8%	−14.4%	3.3%	−23.8%	−45.1%	−14.8%
PEA	−14.2%	−3.1%	−12.8%	3.7%	−6.2%	−9.1%
THC	−3.3%	−0.5%	−3.2%	−1.0%	−0.1%	−1.5%
11COOH-THC	50.1%	−13.9%	76.5%	−6.2%	128.7%	−19.9%
11OH-THC	0.2%	2.3%	−0.1%	−1.9%	3.5%	−3.9%

Values highlighted in light grey are between 20% and 30%. Values highlighted in dark grey are >30%.

**Table 5 biomolecules-13-00383-t005:** Post-preparative stability for steroids (STs) (**A**) and cannabinoids (CBs) (**B**) at the auto-sampler temperature expressed as mean% difference (n = 3) from T0 (T1: 12 h delay; T2: 24 h delay; T3: 36 h delay) at low and high concentration levels [C] (corresponding to CC2, CC8 concentration levels, respectively, of the calibration curve).

(**A**) **STs**						
	**T1**		**T2**		**T3**	
**Analyte**	**Low [C]**	**High [C]**	**Low [C]**	**High [C]**	**Low [C]**	**High [C]**
PREG	12.5%	1.6%	0.2%	0.3%	9.5%	−31.8%
PROG	−20.5%	0.4%	−49.6%	−4.2%	−9.0%	1.5%
17OH-PROG	−14.2%	−8.7%	−5.0%	−6.5%	−9.7%	−4.5%
ALLO	−2.5%	4.0%	2.4%	2.5%	2.8%	4.7%
EPIALLO	8.1%	−4.4%	−5.3%	−2.7%	14.7%	−34.0%
EPIPRAG	−5.7%	−1.3%	6.5%	6.5%	0.2%	14.6%
PRAG	−22.8%	−6.2%	−8.5%	0.9%	−16.3%	9.1%
3α,5α-THDOC	−1.9%	−1.2%	−2.2%	2.0%	−16.3%	−24.2%
DHEA	6.5%	3.0%	6.2%	2.7%	2.1%	4.7%
TESTO	−3.4%	−0.5%	−3.8%	1.6%	−1.3%	−4.9%
DHT	0.0%	0.2%	3.2%	5.5%	−4.9%	2.4%
(**B**) **CBs**						
	**T1**		**T2**		**T3**	
**Analyte**	**Low [C]**	**High [C]**	**Low [C]**	**High [C]**	**Low [C]**	**High [C]**
2-AG	−9.7%	−8.4%	−8.5%	−5.9%	−18.3%	−14.6%
AEA	−4.3%	−2.9%	0.9%	5.2%	−9.4%	−14.6%
OEA	3.8%	−12.1%	26.0%	8.2%	−80.5	−82.2%
PEA	5.1%	−6.8%	10.2%	−3.3%	3.7%	−11.3%
THC	−6.9%	−8.3%	−8.4%	−7.9%	−8.0%	−6.7%
11COOH-THC	9.8%	−10.8%	23.4%	1.5%	24.0%	6.8%
11OH-THC	63.9%	−8.2%	31.9%	−10.8%	50.3%	−4.5%

Values highlighted in light grey are between 20% and 30%, and in dark grey are >30%.

**Table 6 biomolecules-13-00383-t006:** Matrix effect for steroids (STs) (**A**) and cannabinoids (CBs) (**B**) expressed as the% difference at low, mid, and high concentration levels [C] (corresponding to CC2, CC4, and CC8 concentration levels, respectively, of the calibration curve).

(**A**) **STs**			
**Analyte**	**Low [C]**	**Mid [C]**	**High [C]**
PREG	31.2%	15.6%	9.4%
PROG	109.0%	234.0%	23.1%
17OH-PROG	596.8%	−11.2%	−11.2%
ALLO	10.9%	−10.2%	−0.3%
EPIALLO	105.1%	3.3%	13.5%
EPIPRAG	65.8%	50.8%	29.3%
PRAG	106.3%	32.0%	28.9%
3α,5α-THDOC	208.8%	23.2%	−4.7%
DHEA	−97.4%	−5.9%	−15.0%
TESTO	−3.7%	22.4%	10.1%
DHT	40.5%	9.9%	2.4%
(**B**) **CBs**			
**Analyte**	**Low [C]**	**Mid [C]**	**High [C]**
2-AG	10.0%	−3.4%	9.6%
AEA	51.6%	−3.4%	7.0%
OEA	−491.4%	−13.9%	7.8%
PEA	−7.1%	−23.0%	3.8%
THC	19.1%	3.5%	5.2%
11COOH-THC	629.9%	1.9%	−11.6%
11OH-THC	4.3%	−0.7%	8.5%

Values highlighted in light grey are between 20% and 30%, and in dark grey are >30%.

**Table 7 biomolecules-13-00383-t007:** Within- and between-run accuracy for steroids (STs) (**A**) and cannabinoids (CBs) (**B**) at low, mid, and high concentration levels [C] (corresponding to CC2, CC4, and CC8 concentration levels, respectively, of the calibration curve).

(**A**) **STs**					
**Analyte**		**LLOQ [C]**	**Low [C]**	**Mid [C]**	**High [C]**
PREG	Within—Day1	7.3%	8.1%	−0.9%	5.2%
	Within—Day2	−9.1%	18.6%	−0.2%	4.1%
	Within—Day3	5.8%	−2.6%	−1.6%	5.9%
	Between	1.3%	8.0%	−0.9%	5.0%
PROG	Within—Day1	−35.6%	−48.9%	−14.3%	−16.7%
	Within—Day2	−82.8%	−81.8%	−48.6%	−7.7%
	Within—Day3	−82.7%	−24.6%	−27.2%	−14.8%
	Between	−59.2%	−55.9%	−34.5%	−13.1%
17OH-PROG	Within—Day1	37.9%	18.5%	−17.0%	−12.8%
	Within—Day2	6.6%	15.0%	5.7%	−7.6%
	Within—Day3	14.8%	−2.6%	−15.2%	−9.8%
	Between	19.8%	10.3%	−8.8%	−10.1%
ALLO	Within—Day1	−2.8%	9.7%	−9.7%	−0.7%
	Within—Day2	−9.7%	3.6%	−12.4%	−21.7%
	Within—Day3	−14.3%	−5.0%	−12.6%	−12.7%
	Between	−8.9%	2.7%	−11.6%	−11.7%
EPIALLO	Within—Day1	−10.6%	−5.3%	−19.9%	−13.6%
	Within—Day2	−22.6%	−14.9%	−16.6%	−12.9%
	Within—Day3	−15.6%	−55.4%	−24.8%	−13.9%
	Between	−19.6%	−25.2%	−20.4%	−13.5%
PRAG	Within—Day1	−30.8%	−22.3%	−0.4%	−11.5%
	Within—Day2	−3.7%	−2.6%	−7.4%	−9.3%
	Within—Day3	−21.0%	−12.8%	−8.8%	−5.8%
	Between	−21.8%	−12.6%	−5.6%	−8.9%
EPIPRAG	Within—Day1	14.3%	0.0%	−0.2%	7.3%
	Within—Day2	−18.0%	−32.3%	−21.0%	−14.0%
	Within—Day3	−10.9%	−6.9%	−11.9%	−6.5%
	Between	−4.9%	−13.1%	−11.0%	−4.4%
3α,5α-THDOC	Within—Day1	63.8%	26.4%	10.1%	−3.1%
	Within—Day2	80.0%	76.9%	11.2%	1.6%
	Within—Day3	45.4%	31.2%	10.5%	−2.6%
	Between	62.1%	37.9%	10.6%	−1.3%
DHEA	Within—Day1	38.0%	37.4%	−7.7%	−0.5%
	Within—Day2	15.6%	13.3%	−8.7%	−10.7%
	Within—Day3	20.3%	13.4%	−5.4%	−8.6%
	Between	24.6%	21.4%	−7.3%	−6.6%
TESTO	Within—Day1	−48.2%	−46.1%	9.9%	18.2%
	Within—Day2	−48.7%	−41.5%	12.0%	15.3%
	Within—Day3	−5.7%	−17.6%	11.8%	12.1%
	Between	−25.7%	−22.3%	11.3%	15.2%
DHT	Within—Day1	−14.6%	0.1%	−14.2%	−8.4%
	Within—Day2	−21.1%	−20.4%	−23.9%	−20.1%
	Within—Day3	−23.6%	−9.2%	−10.5%	−16.6%
	Between	−23.1%	−9.8%	−16.2%	−15.0%
(**B**) **CBs**					
**Analyte**		**LLOQ**	**Low**	**Mid**	**High**
2-AG	Within—Day1	7.7%	0.9%	3.6%	−2.1%
	Within—Day2	−6.4%	2.8%	−8.6%	−3.5%
	Within—Day3	−2.8%	−4.8%	−8.8%	−6.5%
	Between	−0.5%	−6.4%	−4.6%	−4.1%
AEA	Within—Day1	−4.4%	−1.1%	6.8%	−3.6%
	Within—Day2	−14.5%	−3.2%	−1.5%	−13.7%
	Within—Day3	−5.8%	−0.3%	3.3%	−10.1%
	Between	−8.2%	−1.5%	2.9%	−9.1%
OEA	Within—Day1	20.2%	17.2%	0.6%	−27.1%
	Within—Day2	−52.4%	−57.6%	−48.0%	−56.2%
	Within—Day3	15.1%	44.1%	39.0%	35.5%
	Between	−34.5%	−22.1%	−21.9%	−20.9%
PEA	Within—Day1	15.8%	14.7%	0.2%	−2.6%
	Within—Day2	11.8%	16.1%	−2.0%	−0.9%
	Within—Day3	4.8%	7.5%	−0.7%	−8.4%
	Between	10.8%	12.8%	−0.8%	−0.8%
THC	Within—Day1	22.4%	10.5%	12.8%	11.5%
	Within—Day2	28.3%	23.5%	22.5%	17.3%
	Within—Day3	16.0%	9.9%	7.5%	0.7%
	Between	23.5%	14.6%	15.3%	9.8%
11COOH-THC	Within—Day1	28.8%	24.8%	24.8%	10.1%
	Within—Day2	31.5%	33.2%	23.8%	12.1%
	Within—Day3	26.9%	20.8%	18.3%	8.5%
	Between	28.2	26.7%	22.6%	10.2%
11OH-THC	Within—Day1	−3.1%	−9.3%	−13.3%	−19.9%
	Within—Day2	11.1%	20.3%	0.8%	−13.8%
	Within—Day3	−4.3%	−5.7%	−13.4%	−26.2%
	Between	1.2%	1.8%	−8.6%	−21.3%

Values not highlighted are in the acceptance criteria (±20%). Values highlighted in light gray are between ±30% and in dark gray are not included in the ±30% limits.

**Table 8 biomolecules-13-00383-t008:** Within- and between-run precision for steroids (STs) (**A**) and cannabinoids (CBs) (**B**) at low, mid, and high concentration levels (corresponding to CC2, CC4, and CC8 concentration levels, respectively, of the calibration curve).

(**A**) **STs**					
**Analyte**		**LLOQ [C]**	**Low [C]**	**Mid [C]**	**High [C]**
PREG	Within	18.8%	16.1%	10.2%	6.8%
	Between	1.2%	1.9%	10.2%	6.8%
PROG	Within	67.1%	69.1%	23.1%	11.4%
	Between	21.6%	2.8%	1.7%	0.0%
17OH-PROG	Within	42.8%	40.9%	20.4%	9.6%
	Between	21.6%	40.9%	1.9%	9.6%
ALLO	Within	22.8%	27.6%	14.7%	4.8%
	Between	22.8%	27.6%	14.7%	4.8%
EPIALLO	Within	29.6%	57.4%	17.8%	7.1%
	Between	3.2%	10.6%	17.8%	7.1%
PRAG	Within	48.8%	21.0%	12.2%	8.1%
	Between	5.1%	21.0%	12.2%	8.1%
EPIPRAG	Within	11.2%	27.8%	17.9%	10.2%
	Between	9.9%	7.8%	1.8%	0.8%
3α,5α-THDOC	Within	3.3%	6.7%	6.6%	3.4%
	Between	1.2%	0.8%	6.6%	0.1%
DHEA	Within	17.8%	11.1%	6.3%	1.9%
	Between	5.0%	3.5%	6.3%	0.5%
TESTO	Within	40.8%	30.4%	23.7%	18.4%
	Between	40.8%	30.4%	23.7%	18.4%
DHT	Within	23.8%	10.8%	22.8%	1.8%
	Between	23.8%	10.8%	22.8%	0.7%
(**B**) **CBs**					
**Analyte**		**LLOQ [C]**	**Low [C]**	**Mid [C]**	**High [C]**
2-AG	Within	10.8%	9.7%	8.7%	2.6%
	Between	3.2%	9.7%	3.1%	0.3%
AEA	Within	19.5%	9.7%	5.6%	4.9%
	Between	19.5%	9.7%	1.8%	2.8%
OEA	Within	7.3%	4.4%	9.6%	51.4%
	Between	38.1%	41.6%	29.8%	7.2%
PEA	Within	4.0%	7.2%	4.7%	2.6%
	Between	−0.8%	1.4%	4.7%	1.4%
THC	Within	4.1%	7.2%	2.2%	2.0%
	Between	2.0%	3.7%	3.4%	3.4%
11COOH-THC	Within	5.0%	7.1%	4.5%	3.5%
	Between	5.0%	2.0%	4.5%	0.2%
11OH-THC	Within	5.4%	1.8%	1.8%	4.2%
	Between	3.3%	7.4%	4.1%	2.4%

Values not highlighted are in the acceptance criteria (CV < 20%). Values highlighted in light gray are between 20% < CV < 30%, and in dark gray are >30%.

**Table 9 biomolecules-13-00383-t009:** Solution stability for steroids (STs) (**A**) and cannabinoids (CBs) (**B**) expressed as mean% difference (n = 3) from a fresh solution.

(A) STs				(B) CBs			
Analyte	T0	T1		Analyte	T0	T1	
	WS (4 h at RT)	StS	WS		WS (4 h at RT)	StS	WS
PREG	1.0%	−15.0%	−3.2%	THC	4.8%	−15.4%	−2.0%
PROG	3.8%	−28.2%	−43.6%	11COOH-THC	−5.4%	−9.9%	−0.2%
17OH-PROG	−2.6%	4.3%	20.3%	11OH-THC	−4.6%	4.5%	9.6%
ALLO	9.4%	−23.8%	−5.8%	2-AG	−5.0%	5.9%	23.8%
EPIALLO	−10.7%	−8.5%	2.2%	AEA	−2.9%	−20.2%	−20.1%
EPIPRAG	−21.6%	−6.4%	−24.1%	OEA	16.3%	−30.2%	−24.0%
PRAG	−11.7%	3.1%	−9.9%	PEA	0.9%	−6.4%	15.5%
3α,5α-THDOC	−3.9%	−8.4%	−8.8%				
DHEA	−4.8%	−7.3%	−8.4%				
TESTO	−27.0%	25.5%	11.2%				
DHT	−9.6%	−2.9%	−9.4%				

Values not highlighted are in the acceptance criteria (<20%). Values highlighted in light gray are between 20% and 30%, and in dark gray are >30%.

**Table 10 biomolecules-13-00383-t010:** Freeze and thaw stability for steroids (STs) (**A**) and cannabinoids (CBs) (**B**) expressed as mean% of difference (n = 4) (compared to T0) at mid and high concentration levels [C].

(**A**) **STs**						
	**T1**		**T2**		**T3**	
**Analyte**	**Mid [C]**	**High [C]**	**Mid [C]**	**High [C]**	**Mid [C]**	**High [C]**
PREG	7.3%	14.2%	−27.0%	−3.2%	−50.1%	−22.8%
PROG	27.5%	20.4%	−11.9%	5.5%	−56.2%	−44.3%
17OH-PROG	5.2%	−15.3%	23.9%	−4.9%	−38.8%	−21.9%
ALLO	−1.1%	19.0%	−31.7%	0.0%	−54.8%	−30.1%
EPIALLO	9.2%	10.1%	−25.1%	1.5%	−42.9%	−19.5%
EPIPRAG	2.4%	−10.9%	−25.0%	−14.9%	−60.8%	−45.6%
PRAG	−18.7%	−14.3%	−32.6%	−16.0%	−64.1%	−45.2%
3α,5α-THDOC	−50.7%	27.9%	−59.1%	25.7%	−84.6%	−36.4%
DHEA	−23.2%	5.6%	11.6%	13.9%	−21.6%	25.0%
TESTO	−21.1%	14.1%	−11.8%	−6.7%	−6.2%	18.9%
DHT	−4.6%	−3.9%	−7.1%	−9.2%	−11.2%	−5.2%
(**B**) **CBs**						
	**T1**		**T2**		**T3**	
**Analyte**	**Mid [C]**	**High [C]**	**Mid [C]**	**High [C]**	**Mid [C]**	**High [C]**
2-AG	−13.2%	−3.5%	10.4%	−2.5%	−22.5%	24.8%
AEA	−11.7%	−6.3%	−13.1%	1.9%	−20.9%	−8.8%
OEA	16.8%	22.9%	−43.7%	−23.9%	250.8%	168.6%
PEA	−16.9%	−3.5%	−18.8%	−5.5%	−2.0%	26.8%
THC	−6.7%	1.0%	−10.3%	−1.7%	−5.3%	30.5%
11COOH-THC	−0.6%	−22.1%	18.3%	−8.8%	−6.9%	4.4%
11OH-THC	7.4%	5.0%	7.1%	4.5%	58.1%	−90.5%

Values highlighted in light grey are between 20% and 30%, and in dark grey are >30%. T1: 24 h post-T0; T2: 24 h post-T1; T3:24 h post-T2.

## Data Availability

Not applicable.

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
