# Peer review of "Applied Clinical Tandem Mass Spectrometry-Based Quantification Methods for Lipid-Derived Biomarkers, Steroids and Cannabinoids: Fit-for-Purpose Validation Methods"

_biomolecules, 2023, doi:10.3390/biom13020383_

Round 1

Reviewer 1 Report

A very well written manuscript, well executed, the partial validation planned and executed at a relevant level (for the purpose) compliant with the ICH M10 guideline.

The only real complaint I have is the term Mother Solution which, besides not being gender neutral in this context unfortunately abbreviates to MS (sic). I would prefer Stock Solution

Author Response

Reviewer 1

A very well written manuscript, well executed, the partial validation planned and executed at a relevant level (for the purpose) compliant with the ICH M10 guideline.

The only real complaint I have is the term Mother Solution which, besides not being gender neutral in this context unfortunately abbreviates to MS (sic). I would prefer Stock Solution

Response: We warmly thank Reviewer 1 for his/her expertise and positive comments. We have replaced the term ‘Mother Solution’ with ‘Stock Solution’ throughout the revised manuscript (in red in the text).

Reviewer 2 Report

In their article, authors described the "the development of GC-MS/MS and LC-MS/MS methods for routine quantification of targeted steroids and cannabinoids".

In the Introduction authors pointed out that "work addressed the optimization of mass spectrometric parameters, including ionization and fragmentation, which are one of the most important parameters governing a quantification method", however in the article they describe the validation of these methods. 

In fact, the authors do not describe the tuning of mass spectrometric parameters, leaving this outside the scope of the article. But this step is basic and usual for the creation of the mass-spectrometry method of analysis with high sensitivity.

And although the article is called "Applied clinical tandem Mass Spectrometry-based Quantification Methods for Lipid-derived Biomarkers: Steroids and Cannabinoids", it describes validation, although it is more correct to say an attempt to validate the method of analysis.

In the Introduction at lines 141-144, the authors say that "Our methods achieved good results in terms of sensitivity and specificity and allowed the separation of similar compounds with specific biological functions. These methods are ready-to-use and reproducible, assuring high-throughput performance", although as we could see from the results in is not true.

In the Methods section authors pointed out that "tests were conducted following and adapting the guidelines for bioanalytical method development", but only in results (line 261) do they say that "Acceptability was set at ±20%" and use this error ±20% thought all experiments.

Although accordingly FDA, EMA, and ICH guidelines the  "± 15% error of nominal concentrations; except ± 20% at LLOQ"  is accessible!!!

Authors know it and in Discussion (lines 465-466) say that their methods meet the requirements for GLP-qualified methods, citing a very strange document (reference 51) where there are no exact validation parameters, but only a general discussion.

Why in this case authors choose the ±20%, but not ±50% or even ±100%? Indeed, even in this case, some of the validation parameters are outside this range.

These methods are not validated and cannot be used in the work.

What results will you get if for some analytes the accuracy and precision are 50 or 100%?

This is not acceptable!

In its current state, the article cannot be published

Some minor remarks:

1. lines 195-205. It is not clear what the parameters are for GC-MS, what are for LC-MS. Please, divide it and describe it more accurately.

2. line 207: The selected reaction monitoring is not tandem mass spectrometry.

3.  A characteristic chromatogram with separation of all analytes is required. Possibly in supplementary.

4. The concentrations of working and stock solutions must be indicated.

5. In all tables of results you should add the "%" in the table, and remake the tables since it is not clear which parameter is indicated.

6.  The Calibration curves and linearity section should be moved before the Stability section

7. lines 313-333. Using the methanol matrix instead of plasma or artificial plasma is the wrong idea. You will get incorrect data relative to your real samples.

8. Data about the quantification of real samples should be added. First of all, to understand in what range of the calibration curve are real samples. If they are determined closer to the lower range of the calibration curve, where you have a very large error, then this method of analysis cannot be used at all.

9. line 353. The LLOQ also should  meet the requirement of S/N >10

10.  line 383. Within and between runs Accuracy and Precision should also be tested for  LLOQ 

Author Response

Reviewer 2

We thank Reviewer 2 for his/her expertise and helpful comments in rewriting the manuscript. The revised manuscript has been corrected accordingly (in red in the text).

In their article, authors described the "the development of GC-MS/MS and LC-MS/MS methods for routine quantification of targeted steroids and cannabinoids".

In the Introduction authors pointed out that "work addressed the optimization of mass spectrometric parameters, including ionization and fragmentation, which are one of the most important parameters governing a quantification method", however in the article they describe the validation of these methods.

Response: We agree with Reviewer 2 and the term “optimization” has been replaced by the term “validation” in the revised manuscript.

In fact, the authors do not describe the tuning of mass spectrometric parameters, leaving this outside the scope of the article. But this step is basic and usual for the creation of the mass-spectrometry method of analysis with high sensitivity.

Response: The tuning of mass spectrometric parameters was previously optimized and we have added a paragraph in the revised manuscript in the “Chromatographic and mass spectrometer conditions” section with a more accurate description.

And although the article is called "Applied clinical tandem Mass Spectrometry-based Quantification Methods for Lipid-derived Biomarkers: Steroids and Cannabinoids", it describes validation, although it is more correct to say an attempt to validate the method of analysis.

Response: We didn’t use the term “validation” in the title. The manuscript presents a validation process that includes several parameters/variables that are recommended for analytical studies. Hence, with respect to our data, we conclude that our methods do not strictly comply with GLP-requirements. Consequently, we used the term “qualified” methods instead of “validated” methods. As a result, our fit-for-purpose methods can be used in suitable studies. Thus, we suggest to change the title by: “Applied clinical tandem Mass Spectrometry-based Quantification Methods for Lipid-derived Biomarkers, Steroids and Cannabinoids: fit-for-purpose validation methods”.

In the Introduction at lines 141-144, the authors say that "Our methods achieved good results in terms of sensitivity and specificity and allowed the separation of similar compounds with specific biological functions. These methods are ready-to-use and reproducible, assuring high-throughput performance", although as we could see from the results in is not true.

Response: We have changed this part of the introduction in the revised manuscript: “Our methods allowed the separation of similar compounds with specific biological functions and achieved relatively good results of sensitivity and specificity for some of the targeted compounds. These fit-for purpose methods are ready-to-use for suitable studies.”

In the Methods section authors pointed out that "tests were conducted following and adapting the guidelines for bioanalytical method development", but only in results (line 261) do they say that "Acceptability was set at ±20%" and use this error ±20% thought all experiments. Although accordingly FDA, EMA, and ICH guidelines the "± 15% error of nominal concentrations; except ± 20% at LLOQ" is accessible!!!

Response: In the Methods section of the revised manuscript we have added this sentence: “Precision and accuracy acceptability were established at CV<20% and ±20%, respectively, which can be accepted criteria for mass spectrometry-based quantification of endogenous small molecules (Houghton et al., Bioanalysis, 2009).

Authors know it and in Discussion (lines 465-466) say that their methods meet the requirements for GLP-qualified methods, citing a very strange document (reference 51) where there are no exact validation parameters, but only a general discussion.

Response: We have now added other references: Booth et al., The AAPS Journal, 2015; Cowan, Bioanalysis, 2016; Houghton & Chamberlain, Bioanalysis, 2011).

Why in this case authors choose the ±20%, but not ±50% or even ±100%? Indeed, even in this case, some of the validation parameters are outside this range.

Response: We thank the reviewer for this somewhat "direct and critical" comment. We agree that some % were for some analytes well above the acceptance criteria, as it is usually the case at low concentrations for published methods that also include the quantification of many compounds. This is why our fit of purpose methods are qualified for suitable studies. Our methods provide % accuracy and precision for each target analyte at specific range levels. We can therefore select which analyte can be quantified and under which conditions within ± 20% accuracy and precision.

These methods are not validated and cannot be used in the work.

Response: As mentioned before fit of purpose methods can be qualified and used for suitable studies.

What results will you get if for some analytes the accuracy and precision are 50 or 100%?

Response: We have previously answered about the % accuracy and precision of the methods.

This is not acceptable!

In its current state, the article cannot be published

Response: We hope that the revised manuscript assessing all comments of the three reviewers is now acceptable for publication.

Some minor remarks:

  1. lines 195-205. It is not clear what the parameters are for GC-MS, what are for LC-MS. Please, divide it and describe it more accurately.

Response: As requested, this part has been adjusted in the revised manuscript.

  1. line 207: The selected reaction monitoring is not tandem mass spectrometry.

Response: We agree with Referee 2, and the sentence has been corrected in the revised manuscript.

  1. A characteristic chromatogram with separation of all analytes is required. Possibly in supplementary.

Response: Figures S1 and S2 representing the chromatograms of the target analytes of STs and CBs, respectively, have been added in Supplementary of the revised manuscript.

  1. The concentrations of working and stock solutions must be indicated.

Response: As requested, the concentrations of working and stock solutions have been indicated in Tables S2 and S3 for reference and internal standards, respectively, in Supplementary of the revised manuscript.

  1. In all tables of results you should add the "%" in the table, and remake the tables since it is not clear which parameter is indicated.

Response: As requested, the % has been added for all values in all the tables of results. The labelling of items corresponding to the table columns has been also added in the tables. In addition, the between and within accuracy and precision data have been divided in two tables (Table 7 and Table 8 in the revised version of the manuscript) for better reading and understanding.

  1. The Calibration curves and linearity section should be moved before the Stability section

Response: The calibration curves and linearity section has been moved before the Stability section in the revised manuscript and Figure 3 has been changed accordingly.

  1. lines 313-333. Using the methanol matrix instead of plasma or artificial plasma is the wrong idea. You will get incorrect data relative to your real samples.

Response: We agree that guidelines for bioanalytical method validation recommend calibration curves in matrices similar to biological samples; however, they are mainly focused on exogenous analytes, and in our expertise and others, a large variability in results was obtained for endogenous analytes with plasma matrices (Gachet et al., J. Chromatogr. B, 2015). As stated in several analytical studies, quantitative analyses of endogenous compounds are made difficult by the presence of endogenous analytes (Ji Ji et al., Bioanalysis, 2022; Kaur et al., Bioanalysis 2022). This challenge can be overcome using several approaches of calibration (Jones et al., Bioanalysis, 2012; MacNeill et al., ACS Omega, 2022). The more commonly approach is to use an authentic standard spiked into a surrogate matrix devoid of the target analytes.

Therefore, we decided to perform and compare calibration curves in several matrices (biological/surrogate/neat in methanol). Based on our results, we decided to use calibration curves in methanol for three main reasons. First, it is not straightforward to have appropriate human plasma pools (ideally separate pools in men and women), properly sampled, i.e. within a short time frame (to avoid variations in steroid and endocannabinoid levels following a stressful/challenging condition) and respecting the cold chain between blood and plasma collection (to avoid depletion of endocannabinoids due to rapid action of their degrading enzymes). Hence, calibration curves using the purchased human plasma pool (Dutsher, France) did not meet between and within acceptance criteria, since less than 70% of the calibration time points were accepted according to a ± 20% CV. Surprisingly, this was also the case for the surrogate matrix (PBS containing 6% of BSA) commonly used in analytical studies to mimic plasma matrix. Finally, all the calibration curves in methanol met all acceptance criteria and the analyses of variance showed no differences, in terms of slope and linearity, between the calibration curves in human plasma or in the surrogate matrix and the calibration curves prepared in methanol. This lack of difference fits with recommendation for small molecules quantification in spectrometry analysis by using neat solutions (methanol in our case) (Visconti et al., Analytical Acta, 2023): “To assess the application of neat solutions and/or artificial matrices for quantification purposes, a comparison between the slopes of the calibration curves in surrogate matrices [including neat solutions] and the authentic matrix (standard addition method) should be performed. Even if several statistical tests are available for comparing parallelism of curves, the best known is based on analysis of variance”. Furthermore, no matrix effect was observed in our study. Thus, the parallelism and matrix effect tests were in line with the use of the surrogate matrix approach, as recommended (Kaur et al., Bioanalysis, 2022).

For all these reasons we decided to use methanol for the calibration curves in our analytical studies. Accordingly, organic solvents, such as methanol, has been proposed as a “first surrogate matrix candidate” for analytical methods development for non-polar compounds (Wakamatsu et al., Bioanalysis, 2018).

In addition, in analytical studies in real human samples, quality controls (QC) prepared in the biological (human) matrix area at low, middle and high concentration levels of the calibration levels were added in each run to ensure the validity of our analytical methods. Before each run, we measured also the endogenous levels of each analyte in non-spiked samples that is subtracted from QC values for accuracy and precision evaluation.

  1. Data about the quantification of real samples should be added. First of all, to understand in what range of the calibration curve are real samples. If they are determined closer to the lower range of the calibration curve, where you have a very large error, then this method of analysis cannot be used at all.

Response: The ranges of the calibration curve were chosen to span across anticipated levels of endogenous target analytes in real human samples, with an average around mid-level of the curve. And, if it’s not the case we can adapt anyway with the volume of the samples used for quantification. We have added this precision in the results section on the calibration curves in the revised manuscript. We cannot add real measures of non-published data due to confidentiality. However, data in human using our methods have been cited in the conclusions of the manuscript.

  1. line 353. The LLOQ also should meet the requirement of S/N >10

Response: As requested, this has been added in the revised manuscript.

  1. line 383. Within and between runs Accuracy and Precision should also be tested for LLOQ

Response: As requested, within and between runs Accuracy and Precision data were added in Tables 7 and 8 of the revised manuscript. We also made some corrections to some of the values, as we realized there were some errors.

Reviewer 3 Report

This paper represents a large body of work relating to the development and validation of methods for clinical research, which is considered to be of significant value.

Major criticism

However, this reviewer considers that the methodology may be reliant on the instrumentation used and is thus limited and also the accuracy of the method has not been well described and does not reflect current metrological standards and, at least, may be misleading. This reviewer recommends that the authors carefully review their text and redraft. Some specific comments are presented below.

Please comment/discuss how methanolic standards can adequately reflect "accurate" concentrations in human plasma samples.

Abstract

The abstract claims that the development of GC-MS/MS and LC-MS/MS methods have been described but developed methods only have been presented. Please amend appropriately.

What is meant by "more appropriate for endogenous compounds"? Please elucidate; in what way "more appropriate".

You state that the "methods met the criteria for GLP-qualified methods" yet in the Discussion the authors appropriately state "our methods our not strictly considered as GLP validated methods". Suggest amending the abstract appropriately so as not to mislead the reader.

Suggest changing "parameters" to "variables" in relation to validation throughout. See recommendations of Altman, D. G. and J. M. Bland (1999). "Variables and parameters." Bmj-British Medical Journal 318(7199): 1667-1667.

Suggest changing "parameters" to "conditions" for MS conditions throughout.

Methods

The source and purity of the reference standards should be given.

The source of the QC materials is required.

line 167. The term "control matrix" is mentioned. Please be specific. Do you mean in methanol, surrogate matrix or something else?

line 178. Stating that the use of internal standards "...allows to prevent eventual loss during sample preparation.." is misleading. Better to write "allows adjustment for any losses during sample preparation".

Table 1 could be better expressed in the text if figure 2 showed atom numbering.

line 229. What is the concentration of CC4? Please add.

line 231. What is meant by "acceptable accuracy"? Please be specific.

Table 2. Please state precursor and product ion mass windows.

Please state dwell times used.

Please state how many MRM cycles over each chromatographic peak was used.

Table 3. Please give approximate initial concentrations of Stab samples.

Please state how Stab samples were obtained. Please state ethics approval details and informed consent.

Author Response

Reviewer 3

We thank Reviewer 3 for his/her expertise and helpful comments in rewriting the manuscript. The revised manuscript has been corrected accordingly (in red in the text).

This paper represents a large body of work relating to the development and validation of methods for clinical research, which is considered to be of significant value.

Major criticism

However, this reviewer considers that the methodology may be reliant on the instrumentation used and is thus limited and also the accuracy of the method has not been well described and does not reflect current metrological standards and, at least, may be misleading. This reviewer recommends that the authors carefully review their text and redraft. Some specific comments are presented below.

Please comment/discuss how methanolic standards can adequately reflect "accurate" concentrations in human plasma samples.

Response: We agree that guidelines for bioanalytical method validation recommend calibration curves in matrices similar to biological samples; however, they are mainly focused on exogenous analytes, and in our expertise and others, a large variability in results was obtained for endogenous analytes with plasma matrices (Gachet et al., J. Chromatogr. B, 2015). As stated in several analytical studies, quantitative analyses of endogenous compounds are made difficult by the presence of endogenous analytes (Ji Ji et al., Bioanalysis, 2022; Kaur et al., Bioanalysis 2022). This challenge can be overcome using several approaches of calibration (Jones et al., Bioanalysis, 2012; MacNeill et al., ACS Omega, 2022). The more commonly approach is to use an authentic standard spiked into a surrogate matrix devoid of the target analytes.

Therefore, we decided to perform and compare calibration curves in several matrices (biological/surrogate/neat in methanol). Based on our results, we decided to use calibration curves in methanol for three main reasons. First, it is not straightforward to have appropriate human plasma pools (ideally separate pools in men and women), properly sampled, i.e. within a short time frame (to avoid variations in steroid and endocannabinoid levels following a stressful/challenging condition) and respecting the cold chain between blood and plasma collection (to avoid depletion of endocannabinoids due to rapid action of their degrading enzymes). Hence, calibration curves using the purchased human plasma pool (Dutsher, France) did not meet between and within acceptance criteria, since less than 70% of the calibration time points were accepted according to a ± 20% CV. Surprisingly, this was also the case for the surrogate matrix (PBS containing 6% of BSA) commonly used in analytical studies to mimic plasma matrix. Finally, all the calibration curves in methanol met all acceptance criteria and the analyses of variance showed no differences, in terms of slope and linearity, between the calibration curves in human plasma or in the surrogate matrix and the calibration curves prepared in methanol. This lack of difference fits with recommendation for small molecules quantification in spectrometry analysis by using neat solutions (methanol in our case) (Visconti et al., Analytical Acta, 2023): “To assess the application of neat solutions and/or artificial matrices for quantification purposes, a comparison between the slopes of the calibration curves in surrogate matrices (including neat solutions) and the authentic matrix (standard addition method) should be performed. Even if several statistical tests are available for comparing parallelism of curves, the best known is based on analysis of variance”. Furthermore, no matrix effect was observed in our study. Thus, the parallelism and matrix effect tests were in line with the use of the surrogate matrix approach, as recommended (Kaur et al., Bioanalysis, 2022).

For all these reasons we decided to use methanol for the calibration curves in our analytical studies. Accordingly, organic solvents, such as methanol, has been proposed as a “first surrogate matrix candidate” for analytical methods development for non-polar compounds (Wakamatsu et al., Bioanalysis, 2018).

In addition, in analytical studies in real human samples, quality controls (QC) prepared in the biological (human) matrix area at low, middle and high concentration levels of the calibration levels were added in each run to ensure the validity of our analytical methods. Before each run, we measured also the endogenous levels of each analyte in non-spiked samples that is subtracted from QC values for accuracy and precision evaluation.

Abstract

The abstract claims that the development of GC-MS/MS and LC-MS/MS methods have been described but developed methods only have been presented. Please amend appropriately.

Response: As requested, the text has been amended.

What is meant by "more appropriate for endogenous compounds"? Please elucidate; in what way "more appropriate".

Response: As mentioned before, the calibration curves in methanol met all acceptance criteria, while the calibration in biological and surrogate matrix (PBS containing 6% of BSA) did not. In the text, “more appropriate for endogenous compounds” has been replaced by “more compliant with our quantification of endogenous compounds”.

You state that the "methods met the criteria for GLP-qualified methods" yet in the Discussion the authors appropriately state "our methods our not strictly considered as GLP validated methods". Suggest amending the abstract appropriately so as not to mislead the reader.

Response: The abstract has been revised accordingly: “GLP-qualified rather than GLP-validated methods”.

Suggest changing "parameters" to "variables" in relation to validation throughout. See recommendations of Altman, D. G. and J. M. Bland (1999). "Variables and parameters." Bmj-British Medical Journal 318(7199): 1667-1667. Suggest changing "parameters" to "conditions" for MS conditions throughout.

Response: The term “parameters “has been replaced by “variables” concerning validation and by “conditions “for MS throughout the revised version of the manuscript.

Methods

The source and purity of the reference standards should be given. The source of the QC materials is required.

Response: As requested, these information have been added in the supplementary of the revised manuscript (Table S1). QC samples were from the same source as the reference standards, but different stock solutions were prepared at the same concentrations as the reference standards.

line 167. The term "control matrix" is mentioned. Please be specific. Do you mean in methanol, surrogate matrix or something else?

Response: This referred to methanol. It has been mentioned in the revised version of the manuscript, and “control matrix” has been replaced by “neat matrix”.

line 178. Stating that the use of internal standards "...allows to prevent eventual loss during sample preparation.." is misleading. Better to write "allows adjustment for any losses during sample preparation".

Response: As requested, the text has been amended.

Table 1 could be better expressed in the text if figure 2 showed atom numbering.

Response: The atom numbering is now present in Figure 2.

line 229. What is the concentration of CC4? Please add.

Response: CC4 corresponded to 8 times LLOQ. It is now mentioned in the revised manuscript.

line 231. What is meant by "acceptable accuracy"? Please be specific.

Response: In our case acceptable accuracy means within a 20% CV. It has been added in the text of the revised manuscript.

Table 2. Please state precursor and product ion mass windows. Please state dwell times used. Please state how many MRM cycles over each chromatographic peak was used.

Response: The scan times, dwell times and scan width have been added in the revised manuscript.

Table 3. Please give approximate initial concentrations of Stab samples.

Response: Stab samples were tested at low and high concentrations corresponding to CC2 and CC8 concentration levels, respectively, of the calibration curves. This has been added in Table 3.

Please state how Stab samples were obtained. Please state ethics approval details and informed consent.

Response: As mentioned in the ‘samples preparation’ section, [Stab samples were prepared with working solutions of standard analytes spiked with IS, which were obtained by dilution from separate stock solutions... Stab samples were prepared in commercial plasma (human EDTA-3K plasma; Dutscher, France). The human pool plasma was commercially supplied (Catalogue reference: PLA022; Dutsher, France). It was obtained from Biopredic International (France). No details regarding the ethics and informed content were provided by the supplier. The certificate of analysis stated that Biopredic International complies strictly with the ethical rules for donation and collection of human body products in view of research use only, according to the French Law L.1245-2 CSP. The product was collected from blood male donors in duly authorized blood centers and/or clinical laboratories, who explicitly consented to the use of their blood for scientific research.

Round 2

Reviewer 2 Report

Reviewer 2

We thank Reviewer 2 for his/her expertise and helpful comments in rewriting the manuscript. The revised manuscript has been corrected accordingly (in red in the text).

In their article, authors described the "the development of GC-MS/MS and LC-MS/MS methods for routine quantification of targeted steroids and cannabinoids".

In the Introduction authors pointed out that "work addressed the optimization of mass spectrometric parameters, including ionization and fragmentation, which are one of the most important parameters governing a quantification method", however in the article they describe the validation of these methods.

Response: We agree with Reviewer 2 and the term “optimization” has been replaced by the term “validation” in the revised manuscript.

Excuse me, but you did not replace term “optimization” by the term “validation”:  "...our work addressed the optimisation of mass spectrometric conditions, including ionization and fragmentation, which are important conditions governing a quantification method."... (lines 136-137)

In fact, the authors do not describe the tuning of mass spectrometric parameters, leaving this outside the scope of the article. But this step is basic and usual for the creation of the mass-spectrometry method of analysis with high sensitivity.

Response: The tuning of mass spectrometric parameters was previously optimized and we have added a paragraph in the revised manuscript in the “Chromatographic and mass spectrometer conditions” section with a more accurate description.

OK

And although the article is called "Applied clinical tandem Mass Spectrometry-based Quantification Methods for Lipid-derived Biomarkers: Steroids and Cannabinoids", it describes validation, although it is more correct to say an attempt to validate the method of analysis.

Response: We didn’t use the term “validation” in the title. The manuscript presents a validation process that includes several parameters/variables that are recommended for analytical studies. Hence, with respect to our data, we conclude that our methods do not strictly comply with GLP-requirements. Consequently, we used the term “qualified” methods instead of “validated” methods. As a result, our fit-for-purpose methods can be used in suitable studies. Thus, we suggest to change the title by: “Applied clinical tandem Mass Spectrometry-based Quantification Methods for Lipid-derived Biomarkers, Steroids and Cannabinoids: fit-for-purpose validation methods”.

 OK

In the Introduction at lines 141-144, the authors say that "Our methods achieved good results in terms of sensitivity and specificity and allowed the separation of similar compounds with specific biological functions. These methods are ready-to-use and reproducible, assuring high-throughput performance", although as we could see from the results in is not true.

Response: We have changed this part of the introduction in the revised manuscript: “Our methods allowed the separation of similar compounds with specific biological functions and achieved relatively good results of sensitivity and specificity for some of the targeted compounds. These fit-for purpose methods are ready-to-use for suitable studies.”

OK

In the Methods section authors pointed out that "tests were conducted following and adapting the guidelines for bioanalytical method development", but only in results (line 261) do they say that "Acceptability was set at ±20%" and use this error ±20% thought all experiments. Although accordingly FDA, EMA, and ICH guidelines the "± 15% error of nominal concentrations; except ± 20% at LLOQ" is accessible!!!

Response: In the Methods section of the revised manuscript we have added this sentence: “Precision and accuracy acceptability were established at CV<20% and ±20%, respectively, which can be accepted criteria for mass spectrometry-based quantification of endogenous small molecules (Houghton et al., Bioanalysis, 2009).

Ok, you use the statement of Houghton et al.:In general, the criterion applied in our laboratories, where modulation of biomarker concentrations was unknown, has been that the mean concentration of all VCs at each concentration, including the LLOQ VC, has a %RE less than 20% and a %CV within each set of VCs of less than 20%”.

But in the next paragraph Houghton et al write: “Calibration standards should not deviate more than 15% from the nominal concentration, except at the LLOQ standard, where 20% is acceptable”.

Were the calibrations standards not deviate more than 15% from the nominal concentration in your methods? I am not sure.

Authors know it and in Discussion (lines 465-466) say that their methods meet the requirements for GLP-qualified methods, citing a very strange document (reference 51) where there are no exact validation parameters, but only a general discussion.

Response: We have now added other references: Booth et al., The AAPS Journal, 2015; Cowan, Bioanalysis, 2016; Houghton & Chamberlain, Bioanalysis, 2011).

Ok, it's accepted

Why in this case authors choose the ±20%, but not ±50% or even ±100%? Indeed, even in this case, some of the validation parameters are outside this range.

Response: We thank the reviewer for this somewhat "direct and critical" comment. We agree that some % were for some analytes well above the acceptance criteria, as it is usually the case at low concentrations for published methods that also include the quantification of many compounds. This is why our fit of purpose methods are qualified for suitable studies. Our methods provide % accuracy and precision for each target analyte at specific range levels. We can therefore select which analyte can be quantified and under which conditions within ± 20% accuracy and precision.

Yes, my comment was somewhat "direct and critical", as there are “classical” criteria that apply in method validation. And errors of 50-100% or more look somewhat “defiant”.

Add this explanation of your results to the discussion. And specify the limitations of using the methods you suggested. This will be fair to readers.

Otherwise, some student may accept such measurement errors as the norm and will refer to you as proof of his case.

These methods are not validated and cannot be used in the work.

Response: As mentioned before fit of purpose methods can be qualified and used for suitable studies.

Please indicate the limitations of using the methods

What results will you get if for some analytes the accuracy and precision are 50 or 100%?

Response: We have previously answered about the % accuracy and precision of the methods.

The same comment. Please explicitly state the limitations of using.

This is not acceptable!

In its current state, the article cannot be published

Response: We hope that the revised manuscript assessing all comments of the three reviewers is now acceptable for publication.

 See the last comment.

Some minor remarks:

  1. lines 195-205. It is not clear what the parameters are for GC-MS, what are for LC-MS. Please, divide it and describe it more accurately.

Response: As requested, this part has been adjusted in the revised manuscript.

 OK

  1. line 207: The selected reaction monitoring is not tandem mass spectrometry.

Response: We agree with Referee 2, and the sentence has been corrected in the revised manuscript.

 OK

  1. A characteristic chromatogram with separation of all analytes is required. Possibly in supplementary.

Response: Figures S1 and S2 representing the chromatograms of the target analytes of STs and CBs, respectively, have been added in Supplementary of the revised manuscript.

OK, I see.

  1. The concentrations of working and stock solutions must be indicated.

Response: As requested, the concentrations of working and stock solutions have been indicated in Tables S2 and S3 for reference and internal standards, respectively, in Supplementary of the revised manuscript.

 OK

  1. In all tables of results you should add the "%" in the table, and remake the tables since it is not clear which parameter is indicated.

Response: As requested, the % has been added for all values in all the tables of results. The labelling of items corresponding to the table columns has been also added in the tables. In addition, the between and within accuracy and precision data have been divided in two tables (Table 7 and Table 8 in the revised version of the manuscript) for better reading and understanding.

OK

  1. The Calibration curves and linearity section should be moved before the Stability section

Response: The calibration curves and linearity section has been moved before the Stability section in the revised manuscript and Figure 3 has been changed accordingly.

Ok

  1. lines 313-333. Using the methanol matrix instead of plasma or artificial plasma is the wrong idea. You will get incorrect data relative to your real samples.

Response: We agree that guidelines for bioanalytical method validation recommend calibration curves in matrices similar to biological samples; however, they are mainly focused on exogenous analytes, and in our expertise and others, a large variability in results was obtained for endogenous analytes with plasma matrices (Gachet et al., J. Chromatogr. B, 2015). As stated in several analytical studies, quantitative analyses of endogenous compounds are made difficult by the presence of endogenous analytes (Ji Ji et al., Bioanalysis, 2022; Kaur et al., Bioanalysis 2022). This challenge can be overcome using several approaches of calibration (Jones et al., Bioanalysis, 2012; MacNeill et al., ACS Omega, 2022). The more commonly approach is to use an authentic standard spiked into a surrogate matrix devoid of the target analytes.

Therefore, we decided to perform and compare calibration curves in several matrices (biological/surrogate/neat in methanol). Based on our results, we decided to use calibration curves in methanol for three main reasons. First, it is not straightforward to have appropriate human plasma pools (ideally separate pools in men and women), properly sampled, i.e. within a short time frame (to avoid variations in steroid and endocannabinoid levels following a stressful/challenging condition) and respecting the cold chain between blood and plasma collection (to avoid depletion of endocannabinoids due to rapid action of their degrading enzymes). Hence, calibration curves using the purchased human plasma pool (Dutsher, France) did not meet between and within acceptance criteria, since less than 70% of the calibration time points were accepted according to a ± 20% CV. Surprisingly, this was also the case for the surrogate matrix (PBS containing 6% of BSA) commonly used in analytical studies to mimic plasma matrix. Finally, all the calibration curves in methanol met all acceptance criteria and the analyses of variance showed no differences, in terms of slope and linearity, between the calibration curves in human plasma or in the surrogate matrix and the calibration curves prepared in methanol. This lack of difference fits with recommendation for small molecules quantification in spectrometry analysis by using neat solutions (methanol in our case) (Visconti et al., Analytical Acta, 2023): “To assess the application of neat solutions and/or artificial matrices for quantification purposes, a comparison between the slopes of the calibration curves in surrogate matrices [including neat solutions] and the authentic matrix (standard addition method) should be performed. Even if several statistical tests are available for comparing parallelism of curves, the best known is based on analysis of variance”. Furthermore, no matrix effect was observed in our study. Thus, the parallelism and matrix effect tests were in line with the use of the surrogate matrix approach, as recommended (Kaur et al., Bioanalysis, 2022).

For all these reasons we decided to use methanol for the calibration curves in our analytical studies. Accordingly, organic solvents, such as methanol, has been proposed as a “first surrogate matrix candidate” for analytical methods development for non-polar compounds (Wakamatsu et al., Bioanalysis, 2018).

In addition, in analytical studies in real human samples, quality controls (QC) prepared in the biological (human) matrix area at low, middle and high concentration levels of the calibration levels were added in each run to ensure the validity of our analytical methods. Before each run, we measured also the endogenous levels of each analyte in non-spiked samples that is subtracted from QC values for accuracy and precision evaluation.

Okay, your explanation makes perfect sense.

But then I do not understand why you have such a large measurement error if you use methanol as a matrix.

In my experience, calibration in methanol almost always turns out perfect with minimal deviations.

  1. Data about the quantification of real samples should be added. First of all, to understand in what range of the calibration curve are real samples. If they are determined closer to the lower range of the calibration curve, where you have a very large error, then this method of analysis cannot be used at all.

Response: The ranges of the calibration curve were chosen to span across anticipated levels of endogenous target analytes in real human samples, with an average around mid-level of the curve. And, if it’s not the case we can adapt anyway with the volume of the samples used for quantification. We have added this precision in the results section on the calibration curves in the revised manuscript. We cannot add real measures of non-published data due to confidentiality. However, data in human using our methods have been cited in the conclusions of the manuscript.

 It's not exactly what I wanted to see, but ok

  1. line 353. The LLOQ also should meet the requirement of S/N >10

Response: As requested, this has been added in the revised manuscript.

 Ok

  1. line 383. Within and between runs Accuracy and Precision should also be tested for LLOQ

Response: As requested, within and between runs Accuracy and Precision data were added in Tables 7 and 8 of the revised manuscript. We also made some corrections to some of the values, as we realized there were some errors.

OK

Table 3. Please check "Units". Typically, a concentration is used for the calibration curve.

The authors have significantly improved the manuscript.

Although I still have a negative attitude towards such large errors in measurements, and the publication of such data, if a limitation on the use of methods will be explicitly indicated, this will be acceptable.

I leave the decision on acceptance for publication to the editor.

Author Response

We thank Reviewer 2 for his/her expertise and helpful comments in rewriting the manuscript. The revised manuscript has been corrected accordingly (in red in the text).

Also, we have now addressed the additional comments below in the version 2 of the revised manuscript (in green in the text).

In their article, authors described the "the development of GC-MS/MS and LC-MS/MS methods for routine quantification of targeted steroids and cannabinoids".

In the Introduction authors pointed out that "work addressed the optimization of mass spectrometric parameters, including ionization and fragmentation, which are one of the most important parameters governing a quantification method", however in the article they describe the validation of these methods.

Response: We agree with Reviewer 2 and the term “optimization” has been replaced by the term “validation” in the revised manuscript.

Excuse me, but you did not replace term “optimization” by the term “validation”:  "...our work addressed the optimisation of mass spectrometric conditions, including ionization and fragmentation, which are important conditions governing a quantification method."... (lines 136-137)

Response: We apologize for this omission. The term ‘optimisation’ has been now replaced by the term ‘validation’ in the version 2 of the revised manuscript.

In fact, the authors do not describe the tuning of mass spectrometric parameters, leaving this outside the scope of the article. But this step is basic and usual for the creation of the mass-spectrometry method of analysis with high sensitivity.

Response: The tuning of mass spectrometric parameters was previously optimized and we have added a paragraph in the revised manuscript in the “Chromatographic and mass spectrometer conditions” section with a more accurate description.

OK

And although the article is called "Applied clinical tandem Mass Spectrometry-based Quantification Methods for Lipid-derived Biomarkers: Steroids and Cannabinoids", it describes validation, although it is more correct to say an attempt to validate the method of analysis.

Response: We didn’t use the term “validation” in the title. The manuscript presents a validation process that includes several parameters/variables that are recommended for analytical studies. Hence, with respect to our data, we conclude that our methods do not strictly comply with GLP-requirements. Consequently, we used the term “qualified” methods instead of “validated” methods. As a result, our fit-for-purpose methods can be used in suitable studies. Thus, we suggest to change the title by: “Applied clinical tandem Mass Spectrometry-based Quantification Methods for Lipid-derived Biomarkers, Steroids and Cannabinoids: fit-for-purpose validation methods”.

 OK

In the Introduction at lines 141-144, the authors say that "Our methods achieved good results in terms of sensitivity and specificity and allowed the separation of similar compounds with specific biological functions. These methods are ready-to-use and reproducible, assuring high-throughput performance", although as we could see from the results in is not true.

Response: We have changed this part of the introduction in the revised manuscript: “Our methods allowed the separation of similar compounds with specific biological functions and achieved relatively good results of sensitivity and specificity for some of the targeted compounds. These fit-for purpose methods are ready-to-use for suitable studies.”

OK

In the Methods section authors pointed out that "tests were conducted following and adapting the guidelines for bioanalytical method development", but only in results (line 261) do they say that "Acceptability was set at ±20%" and use this error ±20% thought all experiments. Although accordingly FDA, EMA, and ICH guidelines the "± 15% error of nominal concentrations; except ± 20% at LLOQ" is accessible!!!

Response: In the Methods section of the revised manuscript we have added this sentence: “Precision and accuracy acceptability were established at CV<20% and ±20%, respectively, which can be accepted criteria for mass spectrometry-based quantification of endogenous small molecules (Houghton et al., Bioanalysis, 2009).

Ok, you use the statement of Houghton et al.: “In general, the criterion applied in our laboratories, where modulation of biomarker concentrations was unknown, has been that the mean concentration of all VCs at each concentration, including the LLOQ VC, has a %RE less than 20% and a %CV within each set of VCs of less than 20%”.

But in the next paragraph Houghton et al write: “Calibration standards should not deviate more than 15% from the nominal concentration, except at the LLOQ standard, where 20% is acceptable”.

Were the calibrations standards not deviate more than 15% from the nominal concentration in your methods? I am not sure.

Response: In all three calibration curves in methanol, the accuracy of each accepted time point (according to the previously ±20% criterion), was ±15% for both steroids and cannabinoids. In addition, the CV% for all three slopes ranged from 0.20 to 13.30% for steroids and from 7.03 to 14.64% for cannabinoids. Therefore, in the methods and calibration curves sections of the revised manuscript (R2 version), we replace the ±20% criterion with ±15%, which is consistent with the statement of Houghton et al.

Authors know it and in Discussion (lines 465-466) say that their methods meet the requirements for GLP-qualified methods, citing a very strange document (reference 51) where there are no exact validation parameters, but only a general discussion.

Response: We have now added other references: Booth et al., The AAPS Journal, 2015; Cowan, Bioanalysis, 2016; Houghton & Chamberlain, Bioanalysis, 2011).

Ok, it's accepted

Why in this case authors choose the ±20%, but not ±50% or even ±100%? Indeed, even in this case, some of the validation parameters are outside this range.

Response: We thank the reviewer for this somewhat "direct and critical" comment. We agree that some % were for some analytes well above the acceptance criteria, as it is usually the case at low concentrations for published methods that also include the quantification of many compounds. This is why our fit of purpose methods are qualified for suitable studies. Our methods provide % accuracy and precision for each target analyte at specific range levels. We can therefore select which analyte can be quantified and under which conditions within ± 20% accuracy and precision.

Yes, my comment was somewhat "direct and critical", as there are “classical” criteria that apply in method validation. And errors of 50-100% or more look somewhat “defiant”.Add this explanation of your results to the discussion. And specify the limitations of using the methods you suggested. This will be fair to readers.

Otherwise, some student may accept such measurement errors as the norm and will refer to you as proof of his case.

Response: As suggested by referee 2, we have now added an explanation and limitation of our results to the discussion (lines 533-537): "In this regard, it is worth considering the high variability of the results we obtained, especially at low concentration levels for the quantification of specific endogenous compounds in real samples. An additional validation process of our qualified methods would be required for the quantification of these compounds, especially if low concentrations are expected.”

These methods are not validated and cannot be used in the work.

Response: As mentioned before fit of purpose methods can be qualified and used for suitable studies.

Please indicate the limitations of using the methods

Response: see above

What results will you get if for some analytes the accuracy and precision are 50 or 100%?

Response: We have previously answered about the % accuracy and precision of the methods.

The same comment. Please explicitly state the limitations of using.

Response: see above

This is not acceptable!

In its current state, the article cannot be published

Response: We hope that the revised manuscript assessing all comments of the three reviewers is now acceptable for publication.

 See the last comment.

Response: see above

Some minor remarks:

1. lines 195-205. It is not clear what the parameters are for GC-MS, what are for LC-MS. Please, divide it and describe it more accurately.

Response: As requested, this part has been adjusted in the revised manuscript.

 OK

2. line 207: The selected reaction monitoring is not tandem mass spectrometry.

Response: We agree with Referee 2, and the sentence has been corrected in the revised manuscript.

 OK

3. A characteristic chromatogram with separation of all analytes is required. Possibly in supplementary.

Response: Figures S1 and S2 representing the chromatograms of the target analytes of STs and CBs, respectively, have been added in Supplementary of the revised manuscript.

OK, I see.

4. The concentrations of working and stock solutions must be indicated.

Response: As requested, the concentrations of working and stock solutions have been indicated in Tables S2 and S3 for reference and internal standards, respectively, in Supplementary of the revised manuscript.

 OK

5. In all tables of results you should add the "%" in the table, and remake the tables since it is not clear which parameter is indicated.

Response: As requested, the % has been added for all values in all the tables of results. The labelling of items corresponding to the table columns has been also added in the tables. In addition, the between and within accuracy and precision data have been divided in two tables (Table 7 and Table 8 in the revised version of the manuscript) for better reading and understanding.

OK

6. The Calibration curves and linearity section should be moved before the Stability section

Response: The calibration curves and linearity section has been moved before the Stability section in the revised manuscript and Figure 3 has been changed accordingly.

Ok

7. lines 313-333. Using the methanol matrix instead of plasma or artificial plasma is the wrong idea. You will get incorrect data relative to your real samples.

Response: We agree that guidelines for bioanalytical method validation recommend calibration curves in matrices similar to biological samples; however, they are mainly focused on exogenous analytes, and in our expertise and others, a large variability in results was obtained for endogenous analytes with plasma matrices (Gachet et al., J. Chromatogr. B, 2015). As stated in several analytical studies, quantitative analyses of endogenous compounds are made difficult by the presence of endogenous analytes (Ji Ji et al., Bioanalysis, 2022; Kaur et al., Bioanalysis 2022). This challenge can be overcome using several approaches of calibration (Jones et al., Bioanalysis, 2012; MacNeill et al., ACS Omega, 2022). The more commonly approach is to use an authentic standard spiked into a surrogate matrix devoid of the target analytes.

Therefore, we decided to perform and compare calibration curves in several matrices (biological/surrogate/neat in methanol). Based on our results, we decided to use calibration curves in methanol for three main reasons. First, it is not straightforward to have appropriate human plasma pools (ideally separate pools in men and women), properly sampled, i.e. within a short time frame (to avoid variations in steroid and endocannabinoid levels following a stressful/challenging condition) and respecting the cold chain between blood and plasma collection (to avoid depletion of endocannabinoids due to rapid action of their degrading enzymes). Hence, calibration curves using the purchased human plasma pool (Dutsher, France) did not meet between and within acceptance criteria, since less than 70% of the calibration time points were accepted according to a ± 20% CV. Surprisingly, this was also the case for the surrogate matrix (PBS containing 6% of BSA) commonly used in analytical studies to mimic plasma matrix. Finally, all the calibration curves in methanol met all acceptance criteria and the analyses of variance showed no differences, in terms of slope and linearity, between the calibration curves in human plasma or in the surrogate matrix and the calibration curves prepared in methanol. This lack of difference fits with recommendation for small molecules quantification in spectrometry analysis by using neat solutions (methanol in our case) (Visconti et al., Analytical Acta, 2023): “To assess the application of neat solutions and/or artificial matrices for quantification purposes, a comparison between the slopes of the calibration curves in surrogate matrices [including neat solutions] and the authentic matrix (standard addition method) should be performed. Even if several statistical tests are available for comparing parallelism of curves, the best known is based on analysis of variance”. Furthermore, no matrix effect was observed in our study. Thus, the parallelism and matrix effect tests were in line with the use of the surrogate matrix approach, as recommended (Kaur et al., Bioanalysis, 2022).

For all these reasons we decided to use methanol for the calibration curves in our analytical studies. Accordingly, organic solvents, such as methanol, has been proposed as a “first surrogate matrix candidate” for analytical methods development for non-polar compounds (Wakamatsu et al., Bioanalysis, 2018).

In addition, in analytical studies in real human samples, quality controls (QC) prepared in the biological (human) matrix area at low, middle and high concentration levels of the calibration levels were added in each run to ensure the validity of our analytical methods. Before each run, we measured also the endogenous levels of each analyte in non-spiked samples that is subtracted from QC values for accuracy and precision evaluation.

Okay, your explanation makes perfect sense.

But then I do not understand why you have such a large measurement error if you use methanol as a matrix.

In my experience, calibration in methanol almost always turns out perfect with minimal deviations.

Response: As mentioned above, for all calibration curves in methanol (for all steroids and cannabinoids), accuracy was ±15% of the theoretical concentration for 70% of the calibration samples. In addition, the coefficient of correlation (R2) was greater than 0.99, and the slope significantly different from zero (p>0.05). Therefore, our calibration curves met the acceptance criteria described for endogenous quantification of small molecules by mass spectrometry, as outlined by Houghton et al. (Bioanalysis, 2009), as well as in the guidelines for bioanalytical method validation.

8. Data about the quantification of real samples should be added. First of all, to understand in what range of the calibration curve are real samples. If they are determined closer to the lower range of the calibration curve, where you have a very large error, then this method of analysis cannot be used at all.

Response: The ranges of the calibration curve were chosen to span across anticipated levels of endogenous target analytes in real human samples, with an average around mid-level of the curve. And, if it’s not the case we can adapt anyway with the volume of the samples used for quantification. We have added this precision in the results section on the calibration curves in the revised manuscript. We cannot add real measures of non-published data due to confidentiality. However, data in human using our methods have been cited in the conclusions of the manuscript.

 It's not exactly what I wanted to see, but ok

9. line 353. The LLOQ also should meet the requirement of S/N >10

Response: As requested, this has been added in the revised manuscript.

 Ok

10. line 383. Within and between runs Accuracy and Precision should also be tested for LLOQ

Response: As requested, within and between runs Accuracy and Precision data were added in Tables 7 and 8 of the revised manuscript. We also made some corrections to some of the values, as we realized there were some errors.

OK

Table 3. Please check "Units". Typically, a concentration is used for the calibration curve.

Response: The calibration curve was expressed in terms of amounts of reference standards, not concentrations. We have added a sentence explaining why we used absolute amount for the calibration curves. Also, for clarification, we have added the following sentence in the Calibration Curve section: "Levels in the calibration curves were expressed as absolute amounts of reference standards. As such, the volume of the sample during extraction and/or injection can be adjusted for better sensitivity. For quantification, the ratio of peaks was plotted against a corresponding amount of the calibration curve, which was then normalized to the volume of each analysed sample."

The authors have significantly improved the manuscript.

Although I still have a negative attitude towards such large errors in measurements, and the publication of such data, if a limitation on the use of methods will be explicitly indicated, this will be acceptable.

I leave the decision on acceptance for publication to the editor.

Response: We gratefully acknowledge referee 2's comments, which significantly improved the manuscript. As requested, we have added a sentence about the limitations of the reported data.

Reviewer 3 Report

Congratulations on producing such a useful piece of work; this version is much improved. Below is one major and some minor recommended amendments:

Major criticism:

According to Tables 7 and 8, "LLOQ"s are shown but for some analytes, e.g. progesterone, are far outside normally accepted criteria. Thus you should indicate the LLOQ for these analytes that are presumably at much greater concentrations.

Minor comments:

line 183, change IS to read "reference standard" since both normal and heavy labeled standards are included in Table S1 as was requested.

line 219, change "showed" to read "shown" and delete "analysis".

line 222, change "parameters" to read "conditions"

line 225, delete "compounds"

line 245, change "show" to "shown"

line 253, change "Samples" to "Sample"

line 292, change "levels" to "concentrations"

line 273, change "H2O" to "H2O"

line 301 Table 3, change "levels" to "concentrations"

Units not properly specified. Presumably you mean "pg/mL", etc. please clarify. Also please provide conversion factors where pmol has been used to facilitate other scientists following your work making up appropriate concentrations of solutions.

line 347 Table 4

From the text following the table it appears that mean % concentration differences are being tabulated. If so, please add the word "mean" before "% difference". The shading is stated to indicate a CV >20 or 30%. This seems to contradict "% concentration difference". Are you perhaps looking at the scatter of the data rather than % concentration difference? Please clarify.

Please add number of replicates (n=3?) to the legend.

line 375 Table 5. Same comments as for Table 4 above applies.

line 415 Table 5. Same comments as for Table 4 above applies.

line 489 Table 9. Same comments as for Table 4 above applies.

line 505 Table 10. Same comments as for Table 4 above applies.

line 560 Insert "stated" before "criteria"

Author Response

Comments and Suggestions for Authors

Congratulations on producing such a useful piece of work; this version is much improved. Below is one major and some minor recommended amendments:

Response: We thank referee 3 for acknowledging the improvement in the revised version. We have addressed the following recommended amendments in the revised version (R2) (in green in the text).

Major criticism:

According to Tables 7 and 8, "LLOQ"s are shown but for some analytes, e.g. progesterone, are far outside normally accepted criteria. Thus you should indicate the LLOQ for these analytes that are presumably at much greater concentrations.

Response: We agree with Referee 3, the accuracy and precision at LLOQ that do not meet the acceptance criteria (within ±20%) could be at higher concentrations, although the S/N is greater than 10. We have outlined this point in discussion section of the revised version (R2) of the manuscript: “Moreover, the LLOQ of some compounds do not meet acceptance criteria in terms of accuracy and precision, so the values could be higher and determined more thoroughly. This might be due to endogenous levels that are higher than the spiked amount used for the LLOQ”.

Minor comments:

line 183, change IS to read "reference standard" since both normal and heavy labeled standards are included in Table S1 as was requested.

line 219, change "showed" to read "shown" and delete "analysis".

line 222, change "parameters" to read "conditions"

line 225, delete "compounds"

line 245, change "show" to "shown"

line 253, change "Samples" to "Sample"

line 292, change "levels" to "concentrations"

line 273, change "H2O" to "H2O"

Response: we have made all of the corrections related to the above comments in the revised version (R2) of the manuscript.

line 301 Table 3, change "levels" to "concentrations"

Units not properly specified. Presumably you mean "pg/mL", etc. please clarify. Also please provide conversion factors where pmol has been used to facilitate other scientists following your work making up appropriate concentrations of solutions.

Response: The calibration curve was expressed in terms of amounts of reference standards, not concentrations. This has been changed in Table 3. The conversion factor between pmol and pg has been added in the legend of Table 3. Also, for clarification, we have added the following sentence in the Calibration Curve section: "Levels in the calibration curves were expressed as absolute amounts of reference standards. As such, the volume of the sample during extraction and/or injection can be adjusted for better sensitivity. For quantification, the ratio of peaks was plotted against a corresponding amount of the calibration curve, which was then normalized to the volume of each analysed sample."

line 347 Table 4

From the text following the table it appears that mean % concentration differences are being tabulated. If so, please add the word "mean" before "% difference". The shading is stated to indicate a CV >20 or 30%. This seems to contradict "% concentration difference". Are you perhaps looking at the scatter of the data rather than % concentration difference? Please clarify.

Please add number of replicates (n=3?) to the legend.

line 375 Table 5. Same comments as for Table 4 above applies.

line 415 Table 5. Same comments as for Table 4 above applies.

line 489 Table 9. Same comments as for Table 4 above applies.

line 505 Table 10. Same comments as for Table 4 above applies.

Response: We have added mean % difference in the legend of Tables 4, 5, 9 and 10 and in the above cited lines. The number of replicates has been also added in the legend of Tables 4, 5, 9 and 10. Shading in the tables indicates % values, not CV values. This has been corrected in the revised version (R2).

line 560 Insert "stated" before "criteria"

Response: This has been changed in the revised manuscript (R2 version).